# Enhancing Personnel Selection through the Integration of the Entropy Synergy Analysis of Multi-Attribute Decision Making Model: A Novel Approach

**Sideris Kiratsoudis and Vassilis Tsiantos \***

Didactics, Computational and Applied Mathematics and Micromagnetics Lab, Department of Physics, International Hellenic University, Kavala Campus, 65404 Kavala, Greece; dkkyrat@physics.ihu.gr
\* Correspondence: tsianto@physics.ihu.gr; Tel.: +30-2510-462242

**Abstract:** Personnel selection stands as a pivotal component within the domain of human resource management, intrinsically tethered to the quality of the workforce at large. In this research endeavor, we introduce the Entropy Synergy Analysis of Multi-Attribute Decision Making (ES-MADM) model, an innovative framework expressly designed to rationalize and augment the decision-making processes inherent in the evaluation and selection of personnel within corporate entities. The ES-MADM model systematically navigates the complexities of personnel selection by imbuing objectivity into the assessment criteria, thereby facilitating the structured ranking of potential candidates and establishing a discernible selection sequence. Furthermore, it delves into the statistical significance of these criteria, thereby reinforcing the decision-making process's stability. This research conducts a comparative analysis with alternative multicriteria methodologies and employs sensitivity analysis to ascertain the overall efficacy of the ES-MADM model. This scholarly pursuit, through its rigorous approach, furnishes a comprehensive solution to the intricate challenges surrounding personnel selection, thereby championing a systematic, data-driven approach to underpin pivotal decisions in this sphere.

**Keywords:** multi-criteria decision making (MCDM); multi-attribute decision making (MADM); entropy; information theory; personnel selection



## 1. Introduction

The intricate process of personnel selection, integral to the fabric of organizational decision making, involves a multifaceted journey that spans candidate identification, prioritization, and comprehensive assessment tailored to specific job roles. This nuanced evaluation considers an array of competencies, including knowledge, skills, and experience [1]. The strategic alignment of selection methodologies with organizational objectives emerges as a linchpin for augmenting overall performance, given the substantial repercussions that suboptimal hiring decisions can entail [2,3]. Pioneering scholarly contributions, exemplified by the work of Robertson and Smith in 2001, underscore the imperative of discerning suitable individuals within the expansive candidate pool [4].

The considerable investments in recruitment underscore its pivotal significance, factoring in the costs entailed in recruiting, training, and potentially terminating subpar employees [5]. While historical personnel selection methods predominantly relied on experimental and statistical techniques with interviews playing a pivotal role, the exploration of multi-criteria decision-making (MCDM) techniques has been a relatively underexplored terrain [6].

The overarching objective of personnel selection is to judiciously align individuals with suitable job roles, necessitating a formal, systematic, and rational selection model underpinned by meticulously defined criteria [7]. Multi-criteria decision making (MCDM), a salient domain in the decision-making landscape, grapples with the complexity of decision

problems categorized into multi-attribute decision making (MADM) and multi-objective decision making (MODM) [8]. Scholars have harnessed methodologies such as AHP, ANP, TOPSIS, and ES with increasing prominence in personnel selection [1]. Noteworthy approaches including AHP [9], TOPSIS, ELECTRE, and PROMETHEE [3] have garnered substantial credibility [10,11].

Challenges inevitably emerge in the quantification of numerical values for criteria in personnel selection, rooted in the complexities intrinsic to human resource management. This has led researchers to adapt traditional MCDM techniques to accommodate the inherent fuzziness characterizing the selection problem [12]. Afshari et al. (2011) provide a comprehensive analysis of prevailing MCDM approaches in the evaluation of personnel selection problems. Singh and Kumar Malik (2014) succinctly outline the key attributes of the personnel selection problem, emphasizing the suitability and benefits of commonly utilized MCDM techniques [13].

Zavadskas et al. (2012) accentuate the effectiveness of the Analytical Hierarchy Process (AHP), expert evaluation, and the ARAS method in discerning the selection of a project manager within the construction domain. Their work not only underscores the pragmatic utility of amalgamating these MCDM methodologies but also extends the applicability from academic contexts to real-world scenarios [14].

Karabašević et al. (2015) introduce an integrative methodology, employing the Significance Weight Assessment Ratio Analysis (SWARA) method for discerning criteria significance and the Additive Ratio Assessment (ARAS) method for conclusive evaluations of sales manager candidates [15].

In the pursuit of accurately defining the importance of criteria for selecting a qualified manager in a health institution and identifying an appropriate candidate, Uslu et al. (2021) propose the application of fuzzy AHP and MULTIMOORA methods. Their evaluation, encompassing eight candidates against 12 criteria, utilizes the MULTIMOORA method based on interview results and commission evaluations [16].

Popovic (2021) advocates for the SWARA method to delineate criteria weights and introduces the Combined Compromise Solution (CoCoSo) method for nuanced candidate ranking. This composite approach not only demonstrates success in personnel selection but also provides a versatile model translatable across diverse business sectors [17].

Danisan et al. (2022) implement practical business procedures, leveraging methodologies such as Weighted Scoring (WS), the Analytical Hierarchy Process (AHP), Technique for Order Preference by Similarity to Ideal Solution (TOPSIS), and Preference Ranking Organization Method for Enrichment Evaluations (PROMETHEE). Their methodological synergy facilitates precise personnel selection within a factory setting [18].

In contrast, König and Markus (2022) conduct a meticulous review of empirical research, with a particular focus on the application of machine learning (ML) approaches in personnel selection. They highlight potential biases inherent in ML models, emphasizing the necessity for further research to validate ML applications within selection processes [19].

In their innovative research, Kanakaris and colleagues (2021) delve into the realm of predicting research collaborations. Employing a mix of knowledge graph integration and natural language processing, their study represents a groundbreaking effort to understand the dynamics of collaborative efforts in academia [20]. Expanding on this, in a subsequent study the team applies these methods to assist project managers in the intricate task of personnel selection, showcasing the versatility of their analytical approach [21].

Concurrently, Goretzko and Israel (2021) contribute a comprehensive examination meticulously dissecting challenges tied to using machine learning (ML) for personnel selection. Covering aspects from defining criteria to ensuring transparency, algorithmic fairness, adapting to changing data conditions, and robust performance evaluation, their work offers nuanced insights into the intricacies of implementing fair and accurate ML-based selection algorithms. This analysis significantly contributes to the ongoing discourse on the ethical and methodological dimensions of employing ML in personnel selection [22,23].

Furthermore, Zhang et al. (2023) embark on a probing inquiry into ML's potential to address subgroup differences in personnel selection. Their study specifically explores strategies to mitigate predictive bias while maintaining model accuracy, providing a nuanced understanding of the complexities involved in managing subgroup differences within the realm of ML applications for personnel selection. This depth contributes valuable insights to ongoing discussions on refining and optimizing ML methodologies in personnel decision making [24].

## 2. Model Framework Overview: ES-MADM Structure

### 2.1. The Contribution of Entropy as a MADM Model Tool in the Personnel Selection Problem

An exploration of the existing literature underscores a noticeable gap in the field of personnel selection. Specifically, there is a lack of a standardized framework for objectively defining the criteria used to assess potential candidates. Additionally, there is a dearth of tools capable of impartially quantifying the significance of these criteria and their collective impact during the intricate evaluation and selection process.

In response to this research gap, the proposed Entropy Synergy Multi-Attribute Decision Making (ES-MADM) model introduces a methodological approach that leverages mathematical tools and concepts rooted in entropy and information theory. Operating as a Multi-Attribute Decision Making (MADM) model, ES-MADM initiates its process by statistically assessing the importance of each criterion. These assessments are not empirically derived but are instead inferred from the inherent information within the data. The model then combines these objective assessments with more subjective criteria weights to produce comprehensive integrated importance weights.

In the subsequent phase, the model employs information theory to subject the alternatives to a statistical evaluation. This process reveals the relative significance of each candidate under consideration for selection, providing decision makers with valuable ranking scores for each individual.

It is worth noting that the ES-MADM model uniquely uncovers the statistical significance of each criterion within the context of the decision-selection process. This empowers decision makers to identify the most crucial criteria and pinpoint critical threshold values for both criteria and candidates. These thresholds can significantly impact candidate rankings and thus influence the overall decision hierarchy.

Lastly, the model employs information theory to statistically compute the overall stability of the decision-making process. This involves quantifying the intricate web of interdependencies among the diverse criteria and the range of alternatives (evaluated personnel). By doing so, ES-MADM offers a nuanced insight into the precision and reliability of the decision-making outcomes.

### 2.2. Basic Concepts in Entropy and Information Theory

We present the core concepts and terminology rooted in information theory, which serve as the foundational framework for the construction of the ES-MADM model. This overview is intended to provide a scholarly introduction to the fundamental structure upon which the model is thoughtfully crafted.

#### 2.2.1. Entropy of an Information Source

Based on the probability of each source symbol to be communicated, the Shannon entropy $\mathcal{S}$, in units of bits (per symbol), is given by the Equation (1) [25].

$$\mathcal{S} = -\sum_i p_i log_2(p_i) \tag{1}$$

where $p_i$ refers to the probability of occurrence of the $i$th possible value of the source symbol-fact-circumstance. It should be noticed that the choice of logarithmic base in the formulae determines the unit of information entropy that is used. A common unit of information is the "bit", based on the binary logarithm $(log_2)$. It is evident that in instances

where pi equals zero ($p_i = 0$), denoting a scenario of zero probability, this results in a state of absolute information certainty, characterized by zero entropy.

### 2.2.2. Joint Entropy

The concept of joint entropy, denoted as S(X,Y), pertains to the entropy associated with the combination of two discrete random variables, X and Y. Essentially, it quantifies the uncertainty or disorder in the joint distribution of (X, Y). When X and Y are independent random variables, their joint entropy can be calculated as the sum of their individual entropies. This mathematical expression, as presented in Equation (2), rigorously formalizes the notion of joint entropy [26].

$$S(X,Y) = -\sum_{x \in X} \sum_{y \in Y} [p(x,y)log_2 p(x,y)] \tag{2}$$

### 2.2.3. Conditional Entropy

The conditional entropy or conditional uncertainty of variable Y given the random variable X (also called as the equivocation of Y about X) is the average conditional entropy over X [27]. The equation for conditional entropy is computed by Equation (3).

$$S(Y|X) = -\sum_{x \in X} \sum_{y \in Y} [p(x,y)log_2 p(y/x)] \tag{3}$$

An essential relationship between joint entropy and conditional entropy lies in the fact that the conditional entropy, denoted as S(Y|X), is a constituent component of the joint entropy, S(Y, X). To be more precise, we can assess the conditional entropy by isolating the entropy of Y, as articulated in Equation (4).

$$S(Y|X) = S(X;Y) - S(Y) \tag{4}$$

### 2.2.4. Mutual Information

Mutual information quantifies the extent to which the knowledge of one random variable can enhance our understanding of another. In simpler terms, it measures how much information about one variable you can gain by observing another [28]. It is important in communication where it can be used to maximize the amount of information shared between sent and received signals. The mutual information of *X,Y* is given by Equation (5).

$$J(X;Y) = \sum_{x \in X} \sum_{y \in Y} \left[ p(x,y)log \frac{p(x,y)}{p(x)p(y)} \right] \tag{5}$$

### *2.3. The ES-MADM Model*

### 2.3.1. Basic Concepts on Decision Making

- *Decision-Making (DM) Process*

Decision making (DM) is a meticulous and systematic process, involving the deliberate selection of a specific course of action from a set of alternatives based on available information and predefined criteria. The decision-making process follows a structured series of steps outlined in Table 1. To enhance clarity and precision, Table 2 provides a comprehensive listing of notations for each crucial concept, facilitating a cohesive understanding of the decision-making framework. This meticulous documentation aims to ensure effective communication and collaboration among decision makers, fostering a shared understanding of the criteria and information essential in the decision-making journey.

**Table 1.** The decision-making process.

| Steps | Analysis |
|---|---|
| STEP 1. | Specify the $N$ alternatives (actions, options, issues) $(Y_1, Y_2, \ldots, Y_N)$ |
| STEP 2. | Specify the $M$ criteria $(X_1, X_2, \ldots, X_M)$ |
| STEP 3. | Specify the data that has to be collected. |
| STEP 4. | Rank the criteria by assigning weights $(x_1, x_2, \ldots, x_M)$ for each criterion. The weight is also called the importance factor. Each weight is a positive number taking into consideration the relevance of the criterion for each alternative. |
| STEP 5. | Rank the alternatives according to the criteria's values. The order is determined as follows. Each criterion evaluates each alternative with a value $\xi_{\mu v}$, ($\mu = 1, 2, \ldots, M$ and $v = 1, 2, \ldots, N$). The value $\xi_{\mu v}$ is the value (real or symbolic) that the criterion $X_\mu$ receives with respect to the alternative $Y_v$. The matrix $\Xi = [\xi_{\mu v}]$ is known as the "Data Matrix". The rows of the Data Matrix are the values of the alternatives for each criterion, while the columns of the Data Matrix are the values of the criteria for each alternative. The transpose $\Xi^T$ of the Data Matrix D is known as Decision Matrix. In practice, the Data Matrix is constructed according to specific statistical procedures. After estimating the weights for each criterion, we proceed in the ranking of each alternative expressed as the non-negative numbers $(y_1, y_2, \ldots, y_N)$. Based on the ranking values $(y_1^a, y_2^a, \ldots, y_N^a)$ according to ranking methodology ($\alpha$), the optimal alternative is selected and implemented. If the methodology for estimating the ranking values is fixed, the ranking values are denoted as $(y_1, y_2, \ldots, y_N)$. |
| STEP 6. | Check the sensitivity of the selected alternative with respect to small changes of the weights of the criteria. Sensitivity analysis is required to validate the model's robustness. Small modifications to the values of the criteria should not significantly alter the MCDM's outcomes. This assumption must be verified for every MCDM model; otherwise, "Chaos" and unreliable conclusions may result. Moreover, we specify the domains for which the model is reliable. |

**Table 2.** The notation of decision-making process.

| Factor | Notation |
|---|---|
| Criteria | $X_\mu$ ($\mu = 1, 2, \ldots, M$) |
| Criteria Weights | $x_\mu$ ($\mu = 1, 2, \ldots, M$) |
| Value of Criterion $X_\mu$ for Alternative $Y_v$ | $\xi_{\mu v}$ ($\mu = 1, 2, \ldots, M$) ($v = 1, 2, \ldots, N$) |
| Alternatives | $Y_v$ ($v = 1, 2, \ldots, N$) |
| Alternatives Ranking (Method a) | $y_v^a$ ($v = 1, 2, \ldots, N$) |
| Data Matrix | $D = [\xi_{\mu v}]$ ($\mu = 1, 2, \ldots, M$) ($v = 1, 2, \ldots, N$) |

- *Decision Maker*

A decision maker is an individual or entity with the responsibility of making critical choices or assessments that impact a specific process, situation, or project. This role can be held by an individual, such as an executive or manager, or by a group, such as a committee or board. The decision maker is tasked with evaluating available information, considering various options, and assessing potential risks and benefits before reaching a final decision [29]. In this specific context, the "Decision Maker" pertains to the individual or personnel team entrusted with the critical task of conducting the evaluation and selection of personnel. This role necessitates a rigorous assessment of available candidates and the ultimate decision on the individuals to be chosen. The decision maker's responsibility is to meticulously appraise candidate qualifications, considering factors such as knowledge, skills, and experience, and then make informed selections based on established criteria. These selections, orchestrated by the decision maker, carry significant implications for the organization, as they directly impact the composition of its workforce. In the realm of personnel selection, the decision maker plays a pivotal role in shaping the organization's human resources, ensuring that the chosen personnel align with the specific job roles and contribute to the achievement of organizational objectives.

- *Alternatives (Candidates-Personnel Selection)*

Alternatives are "different possible courses of action". Depending on the nature of the decision problem, the alternatives may correspond to various types of activities, materials

or equipment, methods, and procedures that are being investigated to determine the most lucrative, given specified performance criteria [30]. In the domain of personnel selection, "Alternatives" encompass a range of distinct potential choices or candidates. The specific nature of the selection problem determines the diversity of these alternatives, which can pertain to various candidates possessing different qualifications, skills, and experiences. The overarching goal is to ascertain the most suitable candidate(s) based on well-defined performance criteria. Within the context of personnel selection, alternatives denote the assortment of candidates available for evaluation and consideration. These candidates exhibit diverse attributes and competencies, and the selection process necessitates a thorough assessment to identify the most appropriate fit for a given job role.

- *Criteria (Evaluation Attributes)*

Criteria are defined as "tools for evaluating and comparing alternatives from the viewpoint of the consequences of their selection" [30]. Multi-Criteria Analysis is carried out by selecting the appropriate criteria each time after attempting to classify the possible alternatives, a classification that is possible using predetermined standards (there is of course, the possibility to follow other classification techniques, such as non-parametric and statistical methods). It is precisely these standards that permit absolute comparisons, whereas in other problem-solving methods, comparisons are relevant because alternatives are compared to one another and not to a standard. In the context of personnel selection, the term "criteria" encompasses a diverse array of essential elements relevant to the evaluation process. These elements encompass a comprehensive range of factors, which may include, though are not restricted to, candidate qualifications, skills, experiences, and other pertinent attributes.

- *Criteria Weights*

Weights in Multi-Attribute Decision Making (MADM) represent the relative importance assigned to each criterion for evaluating alternatives [31]. These weights reflect the decision maker's subjective assessment of the criteria's significance in the decision-making process. Typically, weights are assigned in a gradual manner, with increasing importance given to criteria that are considered more significant. To ensure equitable and impartial evaluations, criteria weights undergo a process of normalization, whereby their collective sum equals one. This normalization procedure ensures that each criterion contributes proportionally to the overall assessment without any preferential treatment. Often expressed as percentages, normalized weights indicate the relative importance of each criterion concerning the total weight attributed to all criteria [29]. In the context of personnel selection, the determination of criteria weights is of paramount importance within MADM, as these weights govern the amalgamation of various criteria in calculating an aggregate evaluation score for each candidate. To faithfully represent the decision-maker's preferences and priorities, the assignment of weights to each criterion necessitates thorough evaluation and justification. Within the framework of personnel selection, the assigned weights signify the extent of importance ascribed to each criterion, encompassing factors like qualifications, skills, experiences, and other pertinent attributes. These weights offer a quantitative depiction of the relative emphasis placed on different facets of the candidate evaluation process.

- *Decision Matrix*

A decision matrix, also known as a criteria matrix, is a tool used in DM to evaluate and prioritize alternatives based on a set of criteria. The matrix typically lists the alternatives $(Y_1, Y_2, \ldots, Y_N)$ as rows and the criteria $(X_1, X_2, \ldots, X_M)$ as columns, and each cell $\xi_{\nu\mu} \geq 0$ in the matrix represents the value of the alternative $Y_\nu$ with respect to the criterion $X_\mu$ [32]. The use of decision matrices in MADM methods is the most common way to process data and apply a MADM method.

### 2.3.2. Computation Steps of the ES-MADM method

The proposed method is a novel application that address the MDMP, integrating the Entropy Weighting Method, objectifies the weight of the criteria, and is developed in the following steps.

- STEP 1. Define Alternatives-Criteria-Data Matrix

Step 1.A. Define Alternatives and Criteria: During this phase, the problem is thoroughly analyzed and described. The objective is specified, and several alternatives $Y_N$ are developed. Each of these alternatives is characterized by certain attributes-criteria $X_M$.

Step 1.B. Define the Data: The data correspond to the crucial information that has to be obtained and clarified. These pieces of information which may regard the availability of resources, personnel, equipment, cost etc., for each alternative, are those that constitute the criteria $X_\mu$ of the alternative $Y_\nu$.

Step 1.C. Form the Data Matrix: A data matrix is formed in accordance with the selected criteria $X_\mu$ that should correspond to the crucial information that has to be obtained and clarified. Subjective weights $x_\mu{}^{SBJ}$ of importance are given by the experts for each one of the criteria.

- STEP 2. Compute the Conditional Probabilities

The dependence of the actions-decisions-alternatives $Y_\nu$, $\nu = 1, 2, \ldots, N$ from the criteria $X_\mu$, $\mu = 1, 2, \ldots, M$ is described by the conditional probabilities $P(Y = Y_\nu | X = X_\mu)$ of the actions-decisions-alternatives $(Y)$ with respect to the criteria $(X)$. We denote by $\xi_{\mu\nu}$, $\mu = 1, 2, \ldots, M$, $\nu = 1, 2, \ldots, N$, the value of the alternative $Y_\nu$ with respect to the criterion $X_\mu$. From data matrix $(\xi_{\mu\nu})_{MxN}$, conditional probabilities are constructed as in Equation (6).

$$P(Y = Y_\nu | X = X_\mu) = \xi_{\mu\nu} / \sum_{\nu=1}^{N} \xi_{\mu\nu} \quad \mu = 1, 2, \ldots, M , \nu = 1, 2, \ldots, N \quad (6)$$

- STEP 3. *Compute the Integrated Criteria Weights*

Step 3.A. *Normalize the Decision Matrix*: The normalization of the data matrix is a pivotal step aimed at rectifying disparities arising from diverse measurement units and scales. This procedure serves the purpose of converting the distinct scales and units associated with various criteria into a uniform and standardized measurement framework. Specifically, Equation (7) is invoked to effectuate the normalization of the $\xi_{\mu\nu}$ elements within the data matrix, corresponding to the criterion values for each alternative. This approach underscores the formal and concrete nature of the normalization process in data analysis.

$$\rho_{\mu\nu} = \xi_{\mu\nu} / \sum_{\nu=1}^{N} \xi_{\mu\nu} , \ \forall \ \nu = 1, 2, \ldots, N, \quad \mu = 1, 2, \ldots, M \quad (7)$$

Step 3.B. *Compute the Normalized Entropy of each Criterion*: The normalized entropy $h_\mu$ for each criterion $X_\mu$ ($0 \leq h_\mu \leq 1$) is computed as Equation (8).

$$h_\mu = -\left(\frac{1}{\ln N}\right) \sum_{\nu=1}^{N} \rho_{\mu\nu} \ln(\rho_{\mu\nu}), \mu = 1, 2, \ldots, M \quad (8)$$

Step 3.C. *Compute the Diversification Degree*: The measure of information diversification, denoted as $d_\mu$, pertaining to a decision criterion $X_\mu$, finds its expression in Equation (9). When the dissimilarity value reaches 1, it signifies that the outcomes associated with condition $\mu$ exhibit a high degree of diversity and unpredictability, consequently yielding a heightened level of entropy. Conversely, a dissimilarity value of 0 indicates that the outcomes for condition $\mu$ are markedly similar and foreseeable, resulting in a lower level

of entropy. This delineation underscores the formal and concrete relationship between dissimilarity values and entropy in the context of information diversification.

$$d_\mu = 1 - h_\mu, \quad \forall \; \mu = 1, 2, \ldots, M \tag{9}$$

Step 3.D. *Compute the Objective Criteria Weight*: The objective criteria weight is defined by Equation (10). Criteria that feature similar values for all alternatives (Alternatives), should give low values, regarding their diversification degree $d_\mu$.

$$x_\mu^{OBJ} = d_\mu / \sum_{\mu=1}^{M} d_\mu, \qquad \forall \; \mu = 1, 2, \ldots, M \tag{10}$$

Step 3.E. *Compute the Integrated Criteria Weight*: To ascertain the weights of quantitative criteria, especially in scenarios where partial decision preferences are discerned, the utilization of an entropy module has been advocated [33]. This module facilitates a comprehensive evaluation by harmonizing the subjective criteria weights, denoted as $x_\mu^{SBJ}$, and derived from the decision maker (referred to as HQ), with the objective weights represented as $x_\mu^{OBJ}$. This amalgamation is achieved through the calculation of an integrated weight, denoted as $x_\mu^{INT}$, as explicitly defined in Equation (11). The incorporation of both subjective and objective weights serves to enhance the precision and concreteness of the criteria evaluation process, thereby affording a more robust analytical framework.

$$x_\mu^{INT} = \left( x_\mu^{OBJ} \cdot x_\mu^{SBJ} \right) / \sum_{\mu=1}^{M} \left[ x_\mu^{OBJ} \cdot x_\mu^{SBJ} \right], \; \forall \; \mu = 1, 2, \ldots, M \tag{11}$$

- STEP 4. *Compute the Partial Specific Conditional Entropy of the Alternatives*

The computation of the partial specific conditional entropy, denoted as $\mathcal{S}(Y|X = X_\mu)$, for the alternatives within the context of each criterion $X_\mu$, is undertaken through the application of Equation (12).

$$\mathcal{S}(Y|X = X_\mu) = -\sum_{\nu=1}^{N} P(Y = Y_\nu | X = X_\mu) \log_2 \; P(Y = Y_\nu | X = X_\mu) \tag{12}$$

Additionally, the normalized partial specific conditional entropy, referred to as $\mathcal{I}(Y|X = X_\mu)$, concerning the alternatives Y with respect to each specific criterion $X_\mu$, is derived using Equation (13). This process aims to provide a formal and precise means of quantifying the conditional entropy associated with individual criteria and subsequently normalizing it for analytical purposes.

$$\mathcal{I}(Y|X = X_\mu) = \frac{\mathcal{S}(Y|X = X_\mu)}{\log_2 \; N} \tag{13}$$

- STEP 5. *Compute the Entropy of the Alternatives*

The entropy $\mathcal{S}(Y)$ of all alternatives $Y$ is computed by Equation (14) and will be used in Step 6 in order to normalize the overall conditional entropy of the alternatives $Y$ with respect to the criteria $X$:

$$\mathcal{S}(Y) = -\sum_{\nu=1}^{N} P_\nu^Y \log_2 \; P_\nu^Y \tag{14}$$

The computation of $P_\nu^Y$, as expressed in Equation (15), represents the collective probability score assigned to each alternative. This score serves as the basis for ranking the alternatives. By employing this calculation, we can obtain a precise, formal, and definitive

assessment of the likelihood associated with each alternative, enabling a more objective comparison and selection process.

$$P_\nu^Y = \sum_{\mu=1}^{M} P_{\mu\nu} = \sum_{\mu=1}^{M} P(Y = Y_\nu | X = X_\mu) \cdot P_\mu^X \tag{15}$$

where $P_\mu^X$, $\mu = 1, 2, \ldots, M$ corresponds to the global weights (subjective, objective, or integrated). For the integrated weights $P_\mu^X {x_\mu}^{INT}$.

- STEP 6. *Compute the Conditional Entropy of the Alternatives*

The conditional entropy, denoted as $\mathcal{S}(Y|X)$, quantifies the level of dependence of the alternative $Y$ on the criteria $X$. This measure, calculated according to Equation (16), provides valuable insights into the overall decision stability. It assesses how strongly the alternatives rely on the criteria being utilized, allowing for a more rigorous evaluation of their suitability. By considering $\mathcal{S}(Y|X)$, we can obtain a formal and concrete measurement of the extent to which the alternatives are influenced by the specific criteria employed, thereby facilitating a more robust decision-making process.

$$\mathcal{S}(Y|X) = \sum_{\mu=1}^{M} P_\mu^X \mathcal{S}(Y|X = X_\mu) \tag{16}$$

The normalized conditional entropy $\mathcal{I}(Y|X)$ of the alternatives $Y$ with respect to the criteria $X$ is computed as below:

$$\mathcal{I}(Y|X) = \frac{\mathcal{S}(Y|X)}{\mathcal{S}(Y)} \tag{17}$$

The overall computation process and the respective steps are summarized in an indicative example illustrated in Figure 1, where the output results, related to the importance of the criteria, the overall rank of the alternatives, and the overall stability of the decision problem are depicted in yellow frames.

2.3.3. Integrating the ES-MADM Model into the Personnel Selection Problem

The ES-MADM model can be seamlessly integrated into the framework of effective personnel selection, aligning with the principles outlined in the initial paragraph. The computational procedures of the ES-MADM method, extensively examined in Section 2.3.2, offer a lucid and structured approach for decision makers in the context of personnel selection. Figure 2 serves as a visual representation of the comprehensive process, illustrating the seamless integration of the ES-MADM model into the personnel selection decision-making context.

Furthermore, Figure 3 provides a detailed breakdown of the computational steps inherent to the ES-MADM method within the personnel selection process. Table 3 provides a concise summary of the computation steps used, while Table 4 offers a comprehensive analysis of input and output data relevant to the ES-MADM model. This facilitates a detailed understanding of the sequential steps involved in the model.

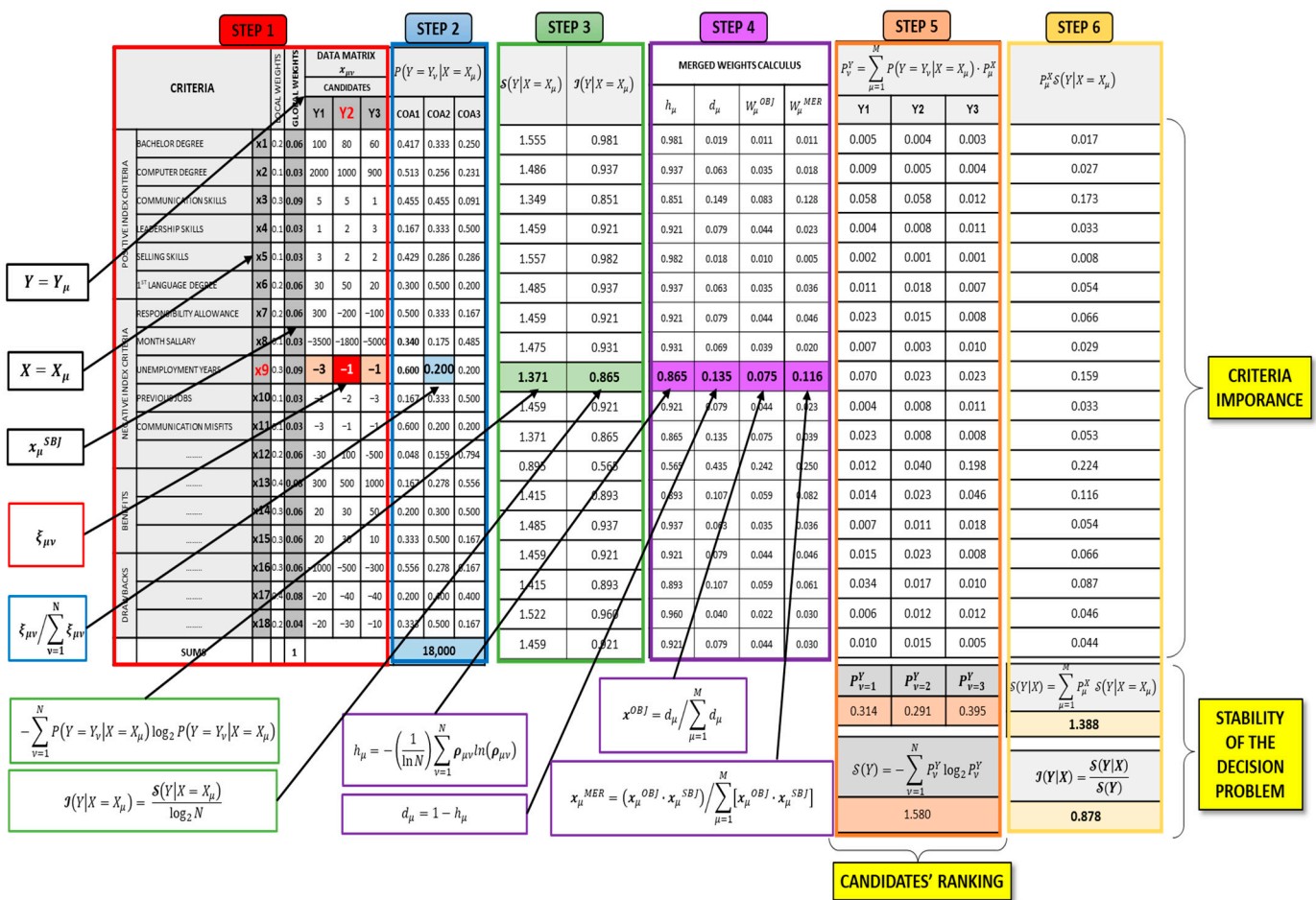

**Figure 1.** Indicative example of the computation steps of the ES-MADM model.

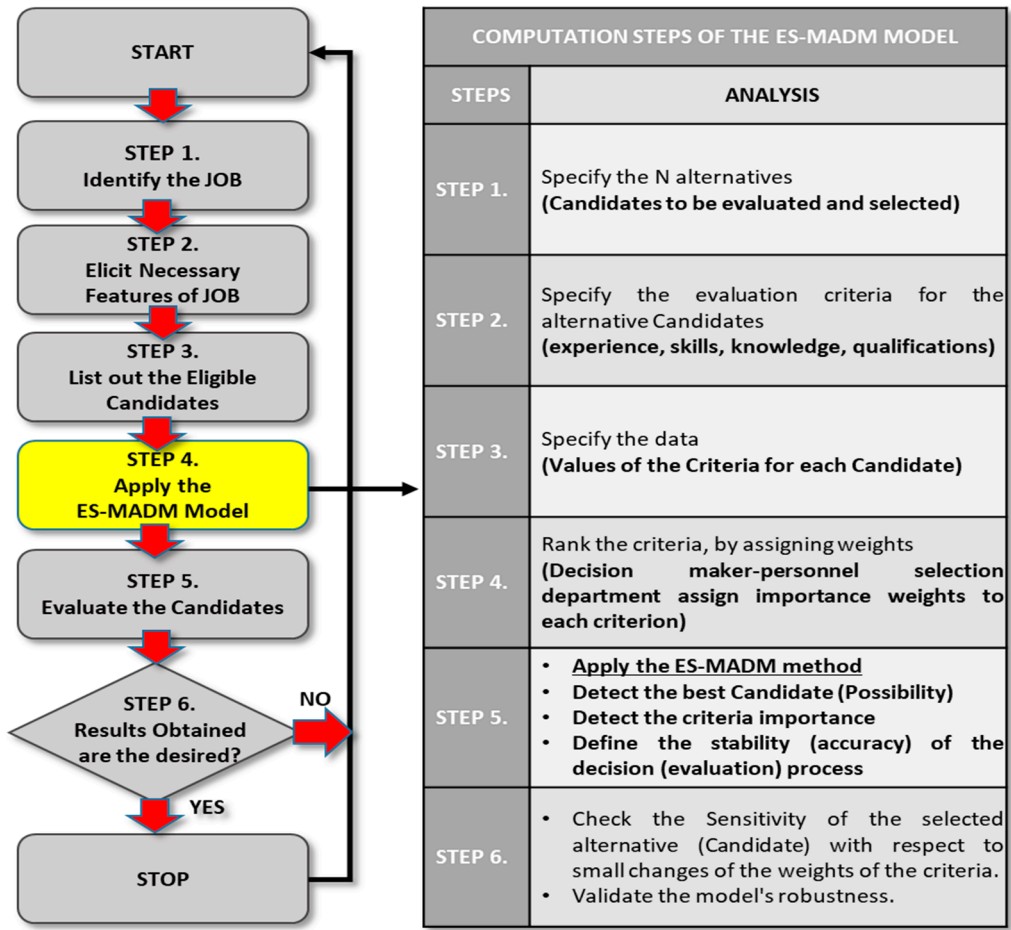

**Figure 2.** Proposed integration of ES-MADM model in the personnel selection problem.

**Figure 3.** Analysis of ES-MADM steps in the context of the personnel selection problem.

**Table 3.** Calculation steps of the ES-MADM model.

| | Substeps | Remarks |
|---|---|---|
| **STEP 1** | | |
| 1.A | Define Alternatives $Y_\nu (\nu = 1, 2, \ldots, N)$ and Criteria $X_\mu (\mu = 1, 2, \ldots, M)$ | Alternatives $Y_\nu$ and criteria $X_\mu$ defined from decision maker (CRITERIA-ALTERNATIVES) |
| 1.B | Data Selection | Data are assembled $\xi_{\mu\nu}$ (values that each $X_\mu$ takes for each $Y_\nu$) (ELEMENTS OF DATA MATRIX) |
| 1.C | Data Matrix $[\Xi_{\mu\nu}]$– Assign Subjective (SBJ) Criteria Weights $x_\mu^{SBJ}$ | Data matrix is completed with data ($\xi_{\mu\nu}$) and $x_\mu^{SBJ}$ (SBJ criteria weights) assigned. (SUBJECTIVE CRITERIA WEIGHTS) |
| **STEP 2** | | |
| 2.A | Compute the Conditional Probabilities $P(Y = Y_\nu \| X = X_\mu)$ | Conditional probability that Y takes the value $Y_\nu$, given that X has the value $X_\mu$. |
| **STEP 3** | | |
| 3.A | Normalize into $[\rho_{\mu\nu}]$ the Data Matrix $[\Xi_{\mu\nu}]$ | Normalization of $\xi_{\mu\nu}$ data matrix elements. Essentially the results are the same with 2. A |
| 3.B | Compute the Normalized Entropy $h_\mu$ of each Criterion | Quantifies the average amount of information or uncertainty associated with the outcome Y for a specific condition $X = X_\mu$ |
| 3.C | Compute the Diversification Degree $d_\mu$ for each criterion | Quantifies the degree of dissimilarity or distinctiveness associated with the specific condition $\mu$ |
| 3.D | Compute the Objective Weight $x_\mu^{OBJ}$ for each criterion | Computation of the objective weights of criteria. (OBJECTIVE CRITERIA WEIGHTS) |
| 3.E | Compute the Integrated Weights $x_\mu^{INT}$ for each criterion | Computation of the integrated weight weights of criteria. (INTEGRATED CRITERIA WEIGHTS) |
| **STEP 4** | | |
| 4.A | Compute the Conditional Entropy $\mathcal{S}(Y\|X = X_\mu)$ of the COAs and the Normalised Conditional Entropy $\mathcal{I}(Y\|X = X_\mu)$ | Calculates the conditional entropy of the variable Y given that X takes on the specific value $X = X_\mu$. It measures the average amount of uncertainty or information required to describe the outcome of Y, considering given value X. |
| | | Computes the mutual information between variables Y and X, given that X takes a specific value $X = X_\mu$. Quantifies the reduction in uncertainty or shared information between Y and X, normalized by the logarithm base 2 of the number of possible outcomes N. (OBJECTIVE CRITERIA SIGNIFICANCE) |
| **STEP 5** | | |
| 5.A | Compute the overall Entropy of the COAs $\mathcal{S}(Y)$ and the overall Score for each COA $P_\nu^Y$ | $\mathcal{S}(Y)$ calculates the entropy of the variable Y, which measures the average amount of uncertainty or information required to describe the possible outcomes of Y |
| | | $P_{\mu\nu}$ combines the conditional probability with the significance of the criterion to evaluate the joint probability or significance of $Y = Y_\nu$ occurring in conjunction with $X = X_\mu$. |
| | | $P_\nu^Y$ computes the probability of the outcome Y taking the value $Y_\nu$. (ALTERNATIVE PROBABILITY-FINAL SCORE) |
| **STEP 6** | | |
| 6.A | Compute the Conditional Entropy $\mathcal{S}(Y\|X)$ of the COAs and the Normalised Conditional Entropy $\mathcal{I}(Y\|X)$ | $\mathcal{S}(Y\|X)$ measures the average amount of uncertainty or information required to describe the outcome of Y, considering the different values of X and their corresponding conditional entropies. $P_\mu^X \mathcal{S}(Y\|X = X_\mu)$ captures the overall weighted conditional entropy for the specific condition $X = X_\mu$. (INTEGRATED CRITERIA SIGNIFICANCE) |
| | | $\mathcal{I}(Y\|X)$ quantifies the reduction in uncertainty about Y when X is known, relative to the total uncertainty in Y. (TOTAL PROBLEM UNCERTAINTY) |

**Table 4.** Summary of the input-output data in ES-MADM model.

| Concept | Nomenclature | Type | Analysis | Data Remarks |
|---|---|---|---|---|
| Alternatives | $Y_\nu$ | INPUT | Defined by the decision-maker(s)—personnel selection department | $(\nu = 1, 2, \ldots, N)$ Where N is the number of the candidates to be evaluated |
| Criteria | $X_\mu$ | INPUT | Defined by the decision maker(s)—type of characteristics serving as criteria for the evaluation of the candidates | $(\mu = 1, 2, \ldots, M)$ Where M is the number of the qualifications-characteristics to be evaluated |
| Elements of Data Matrix | $\zeta_{\mu\nu}$ | INPUT | Defined by the decision maker(s)—refers to the values of each criterion for each candidate | $M \times N$ All data regarding all candidates for all criteria |
| Subjective Criteria Weights | $x_\mu{}^{SBJ}$ | INPUT | Defined by the decision maker(s)—refers to the subjective weights assigned to the criteria | $0 \leq x_\mu{}^{SBJ} \leq 1$ The weight of the criterion increases in proportion to its corresponding value. |
| Normalized Conditional Entropy (Specific) | $\mathcal{I}(Y\|X = X_\mu)$ | OUTPUT | Measures the objective criteria significance (significance of each criterion with respect to the candidate selection process) | $0 \leq \mathcal{I}(Y\|X = X_\mu) \leq 1$ Objective dependence between the criteria and the candidate selection. The objective is to minimize the value as much as possible. |
| Product of Conditional Probability with Criteria Weight | $P_\mu^X \mathcal{S}(Y\|X = X_\mu)$ | OUTPUT | Measures the integrated criteria significance (significance of each criterion) | $0 \leq P_\mu^X \mathcal{S}(Y\|X = X_\mu) \leq 1$ Integrated dependence between the criteria and the selection probability (usually combined with the objective significance). The objective is to maximize the value as much as possible. |
| Probability of $Y_\nu$ | $P_\nu^Y$ | OUTPUT | Measures the probability of selecting each candidate | $0 \leq P_\nu^Y \leq 1$ The weight of the selection probability increases in proportion to its corresponding value. The objective is to maximize the value as much as possible for each candidate. |
| Normalized Conditional Entropy | $\mathcal{I}(Y\|X)$ | OUTPUT | Measures the uncertainty of the decision problem (uncertainty between criteria–candidates). | $0 \leq \mathcal{I}(Y\|X) \leq 1$ Dependence between the overall criteria and the overall candidate selection process, which correspond to the stability of the problem. The objective is to minimize the value as much as possible. |

These visual aids collectively provide a clear and detailed depiction of the decision-making process in personnel selection, assisting decision makers in effectively navigating the analysis and evaluation process.

### 3. Exploring ES-MADM Model Performance

*3.1. An Illustrative Case Study for the ES-MADM Model*

The ES-MADM model's effectiveness and robustness in assessing the likelihood of success in complex decision scenarios, such as personnel selection, are rigorously examined through a comprehensive case study. The dataset for this study was sourced from the work of Oya KORKMAZ [34], who employed the TOPSIS model for personnel selection. The primary objective of this case study is to demonstrate the practical applicability of the ES-MADM model in aiding decision makers to evaluate potential outcomes within the context of personnel selection. By conducting a meticulous analysis utilizing empirical data and scenarios from personnel selection cases, this study aims to showcase the model's utility in providing valuable insights and assisting decision makers in assessing the possibilities of achieving success.

3.1.1. Case Study: Problem Statement

In the study conducted by Oya KORKMAZ [34], a selection process was undertaken to assess 20 candidates who had applied for positions as domestic logistics operation personnel at a logistics company in Mersin, operating within the logistics sector. Out of these 20 candidates, 11 were disqualified due to their inability to meet the required criteria. Consequently, only 9 candidates who satisfied the specified criteria were subjected to further evaluation. The weighted data used to evaluate the candidates were collected through interviews conducted by a team of 7 individuals, comprising a regional director and members of the human resources department. The company identified several key qualities sought in the logistics operation personnel they intended to hire. These qualities included: (1) prior experience in logistics, (2) educational background and training, (3) flexibility in work hours and willingness to work overtime, (4) proficiency in MS Office programs, (5) familiarity with package software commonly used in the logistics field (e.g., ERP, SAP, etc.), and (6) possession of recommendation letters. The data matrix utilized for this assessment is presented in Table 5, while the hierarchical structure of the decision problem is visually depicted in Figure 4.

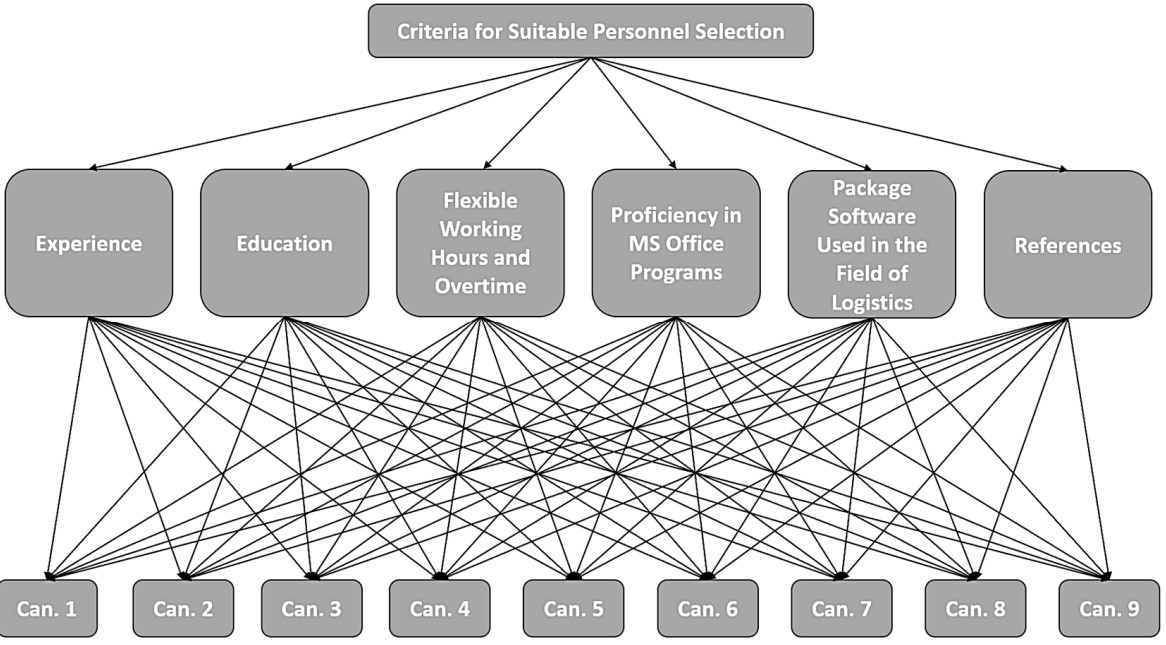

**Figure 4.** Hierarchical structure of the selection process.

**Table 5.** Data matrix for the personnel selection problem (Steps 1 in ES-MADM).

| | Criteria | Weights $x_\mu^{SBJ}$ | Alternatives | | | | | | | | |
|---|---|---|---|---|---|---|---|---|---|---|---|
| | | | C1 | C2 | C3 | C4 | C5 | C6 | C7 | C8 | C9 |
| $X_1$ | Logistics Experience | 0.16667 | 4 | 3 | 3 | 4 | 9 | 4 | 3 | 3 | 3 |
| $X_2$ | Education | 0.16667 | 8 | 8 | 6 | 1 | 5 | 10 | 10 | 7 | 10 |
| $X_3$ | Flexible Working Hours and Overtime | 0.16667 | 5 | 8 | 8 | 10 | 3 | 10 | 6 | 6 | 8 |
| $X_4$ | Proficency in MS Office Programs Package | 0.16667 | 6 | 5 | 7 | 6 | 6 | 7 | 7 | 8 | 8 |
| $X_5$ | Software Used in The Field of Logistics | 0.16667 | 7 | 1 | 1 | 1 | 5 | 8 | 1 | 5 | 6 |
| $X_6$ | Recommendation Letters | 0.16667 | 1 | 1 | 8 | 8 | 1 | 8 | 1 | 1 | 5 |

### 3.1.2. Case Study: ES-MADM (Initial Results)

The entry/input data that correspond to the subjective weight $x_\mu^{SBJ}$ of each criterion and the values $\xi_{\mu\nu}$ of the data matrix were inserted into the ES-MADM model. The model computes the integrated criteria weights $x_\mu^{INT}$ and respectively evaluates the alternative alternatives $P_\nu^Y$, the importance of the criteria $P_\mu^X \mathcal{S}(Y|X = X_\mu)$ and the stability of the decision (evaluation) problem $\mathcal{I}(Y|X)$. All computations are listed in Table 6, while the final results are summarized in Table 7.

The overall results for the winning probabilities $P_\nu^Y$ and the integrated (INT) importance of the criteria $P_\mu^X \mathcal{S}(Y|X = X_\mu)$ are depicted in Figures 5 and 6. Additionally, Figure 7 presents a comparative diagram showcasing the relationship between the integrated importance (INT), the subjective importance $P_\mu^X$, and the objective importance $\mathcal{I}(Y|X = x_\mu)$ of the criteria.

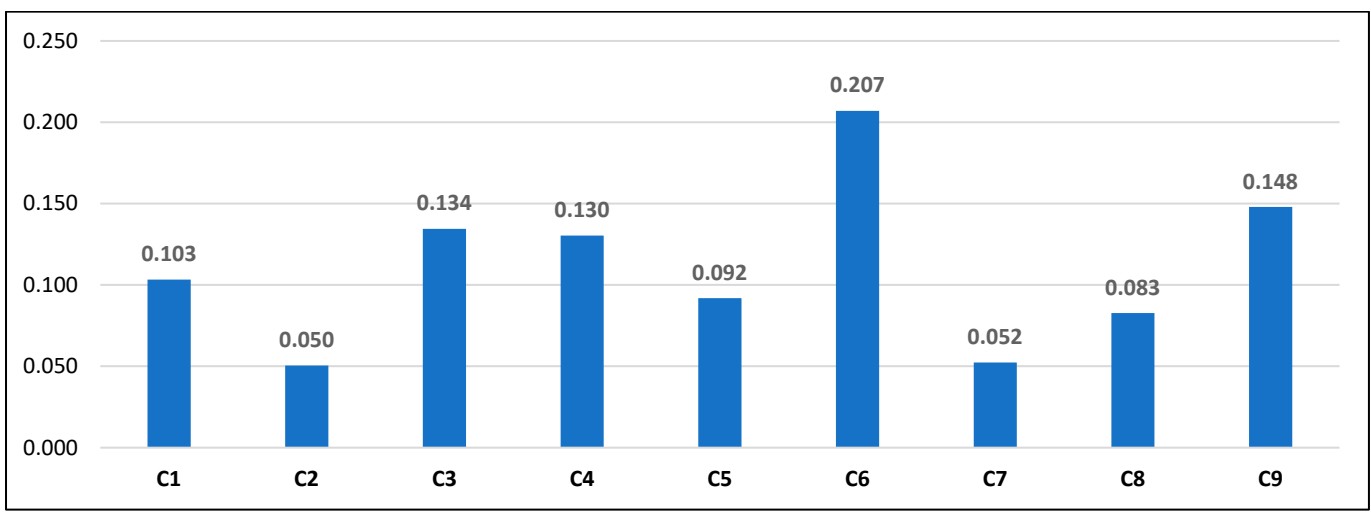

**Figure 5.** Selection probabilities $P_\nu^Y$ for all candidates produced by ES-MADM model.

**Table 6.** Computations summary in ES-MADM model for Steps 2–6.

| Criteria | $P(Y=y_\nu\vert X=x_\mu)=\xi_{\mu\nu}/\sum_{\nu=1}^9 \xi_{\mu\nu}$ | | | | | | | | | $x_\mu^{INT}$ | $\mathcal{I}(Y\vert X=x_\mu)$ | $P_\nu^Y=\sum_{\mu=1}^6 P(Y=y_\nu\vert X=x_\mu)\cdot P_\mu^X$ | | | | | | | | | $P_\mu^X\mathcal{S}(Y\vert X=X_\mu)$ |
|---|---|---|---|---|---|---|---|---|---|---|---|---|---|---|---|---|---|---|---|---|---|
| | C1 | C2 | C3 | C4 | C5 | C6 | C7 | C8 | C9 | | (OBJ Criteria Importance) | C1 | C2 | C3 | C4 | C5 | C6 | C7 | C8 | C9 | (INT Criteria Importance) |
| X1 | 0.111 | 0.083 | 0.083 | 0.111 | 0.250 | 0.111 | 0.083 | 0.083 | 0.083 | 0.093 | 0.962 | 0.010 | 0.008 | 0.008 | 0.010 | 0.023 | 0.010 | 0.008 | 0.008 | 0.008 | 0.284 |
| X2 | 0.123 | 0.123 | 0.092 | 0.015 | 0.077 | 0.154 | 0.154 | 0.108 | 0.154 | 0.108 | 0.956 | 0.013 | 0.013 | 0.010 | 0.002 | 0.008 | 0.017 | 0.017 | 0.012 | 0.017 | 0.328 |
| X3 | 0.078 | 0.125 | 0.125 | 0.156 | 0.047 | 0.156 | 0.094 | 0.094 | 0.125 | 0.057 | 0.977 | 0.004 | 0.007 | 0.007 | 0.009 | 0.003 | 0.009 | 0.005 | 0.005 | 0.007 | 0.177 |
| X4 | 0.100 | 0.083 | 0.117 | 0.100 | 0.100 | 0.117 | 0.117 | 0.133 | 0.133 | 0.011 | 0.995 | 0.001 | 0.001 | 0.001 | 0.001 | 0.001 | 0.001 | 0.001 | 0.002 | 0.002 | 0.036 |
| X5 | 0.200 | 0.029 | 0.029 | 0.029 | 0.143 | 0.229 | 0.029 | 0.143 | 0.171 | 0.308 | 0.876 | 0.062 | 0.009 | 0.009 | 0.009 | 0.044 | 0.070 | 0.009 | 0.044 | 0.053 | 0.854 |
| X6 | 0.029 | 0.029 | 0.235 | 0.235 | 0.029 | 0.235 | 0.029 | 0.029 | 0.147 | 0.422 | 0.829 | 0.012 | 0.012 | 0.099 | 0.099 | 0.012 | 0.099 | 0.012 | 0.012 | 0.062 | 1.110 |

**Table 7.** Summary results table for ES-MADM model.

| Ranking of the Alternatives (Candidates) | | | | | | | | | | Decision Stability | | |
|---|---|---|---|---|---|---|---|---|---|---|---|---|
| $P_{\nu=1}^Y$ | $P_{\nu=2}^Y$ | $P_{\nu=3}^Y$ | $P_{\nu=4}^Y$ | $P_{\nu=5}^Y$ | $P_{\nu=6}^Y$ | $P_{\nu=7}^Y$ | $P_{\nu=8}^Y$ | $P_{\nu=9}^Y$ | $\mathcal{S}(Y\vert X)$ | $\mathcal{S}(Y)=-\sum_{\nu=1}^N P_\nu^Y\log_2 P_\nu^Y$ | $\mathcal{I}(Y\vert X)=\frac{\mathcal{S}(Y\vert X)}{\mathcal{S}(Y)}$ |
| 0.103 | 0.050 | 0.134 | 0.130 | 0.092 | 0.207 | 0.052 | 0.083 | 0.148 | 2.789 | 3.042 | 0.917 |

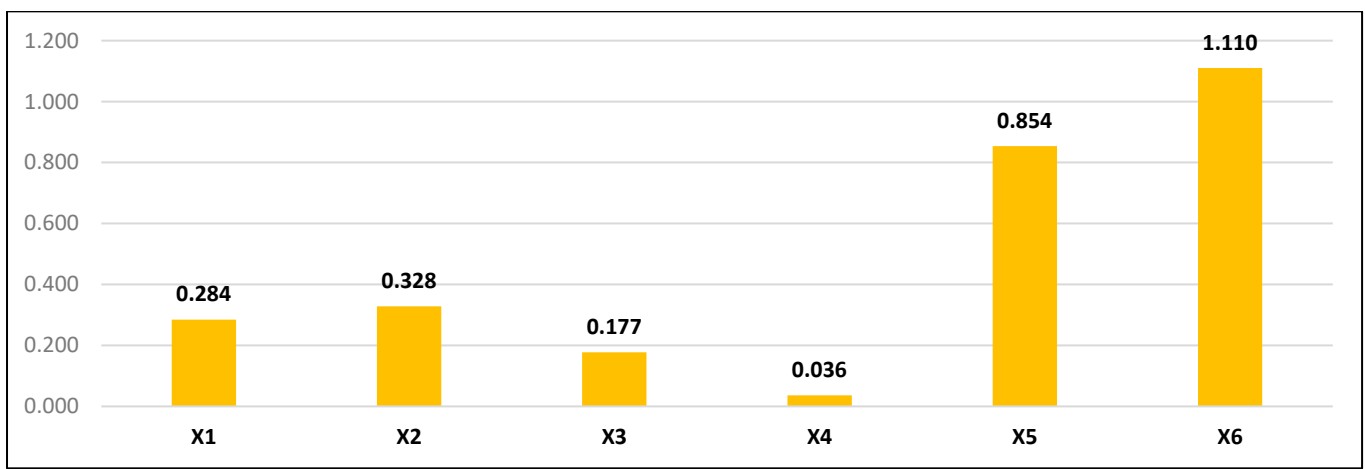

**Figure 6.** Integrated criteria importance $P_\mu^X \mathcal{S}(Y|X = X_\mu)$ produced by ES-MADM model.

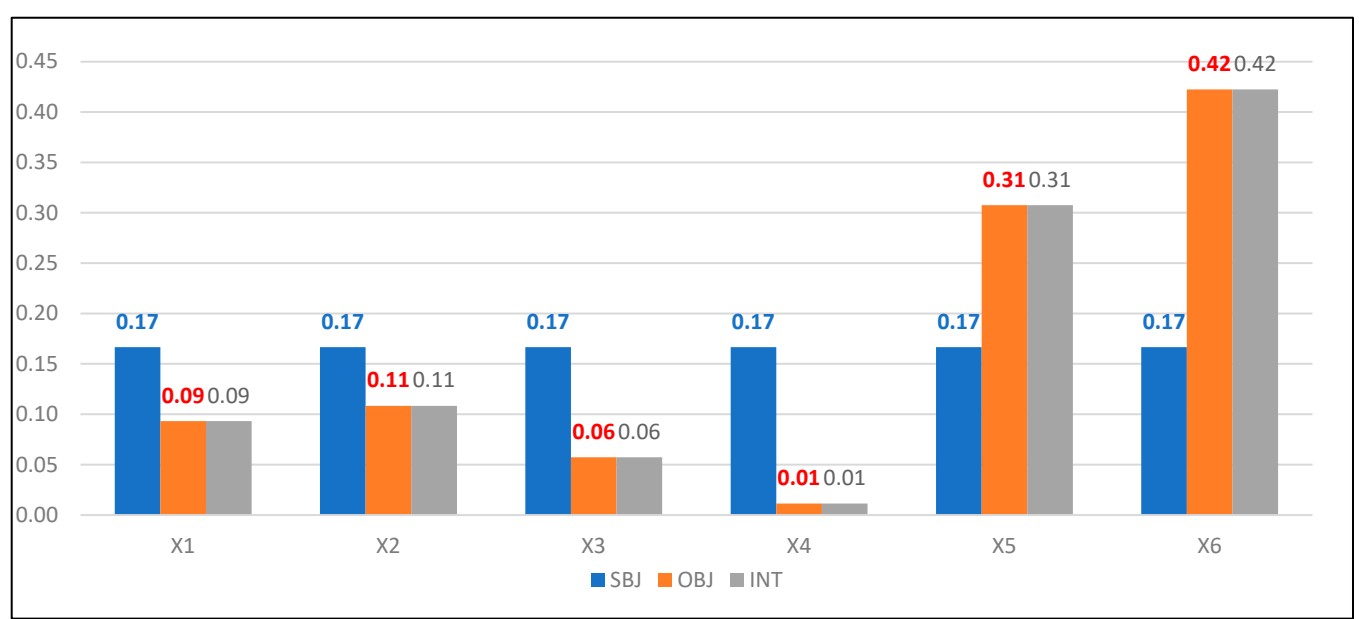

**Figure 7.** Comparison between the values of the integrated (INT), objective (OBJ), and subjective (SBJ) criteria importance.

### 3.1.3. Case Study: ES-MADM (Assigning Non-Equal SBJ Weights to the Criteria)

When decision makers allocate subjective weights ($x_\mu^{SBJ}$) to the criteria that are not equal, the model, as outlined in Equation (11), will produce integrated weight outcomes ($x_\mu^{INT}$). These integrated weights are not solely determined by the objective weight values ($x_\mu^{OBJ}$) associated with the criteria. Instead, they are influenced by both the objective weight values and the subjective weights assigned by decision makers.

This integration occurs by combining the subjective weights with the objective weight values, which are calculated based on the inherent information present in the elements of the data matrix. The subjective weights $x_\mu^{SBJ}$ assigned to the criteria were consolidated and are presented in Table 8, while it is essential to emphasize that the data matrix retains its original content without any alterations.

**Table 8.** Data matrix for the personnel selection problem—non-equal SBJ criteria weights (Step 1 in ES-MADM).

| | Criteria | Weights $x_\mu^{SBJ}$ | Alternatives | | | | | | | | |
|---|---|---|---|---|---|---|---|---|---|---|---|
| | | | C1 | C2 | C3 | C4 | C5 | C6 | C7 | C8 | C9 |
| $X_1$ | Logistics Experience | 0.2 | 4 | 3 | 3 | 4 | 9 | 4 | 3 | 3 | 3 |
| $X_2$ | Education | 0.1 | 8 | 8 | 6 | 1 | 5 | 10 | 10 | 7 | 10 |
| $X_3$ | Flexible Working Hours and Overtime | 0.3 | 5 | 8 | 8 | 10 | 3 | 10 | 6 | 6 | 8 |
| $X_4$ | Proficiency in MS Office Programs | 0.2 | 6 | 5 | 7 | 6 | 6 | 7 | 7 | 8 | 8 |
| $X_5$ | Package Software Used in The Field of Logistics | 0.1 | 7 | 1 | 1 | 1 | 5 | 8 | 1 | 5 | 6 |
| $X_6$ | Recommendation Letters | 0.1 | 1 | 1 | 8 | 8 | 1 | 8 | 1 | 1 | 5 |

The complete set of results produced by the model under this condition is presented in Tables 9 and 10.

The comprehensive findings related to the probabilities of personnel selection $P_\nu^Y$ and the integrated (INT) significance of the criteria are graphically depicted in Figures 8 and 9, respectively. These figures offer a visual representation of the scenario in which non-uniform subjective weights have been assigned to the criteria.

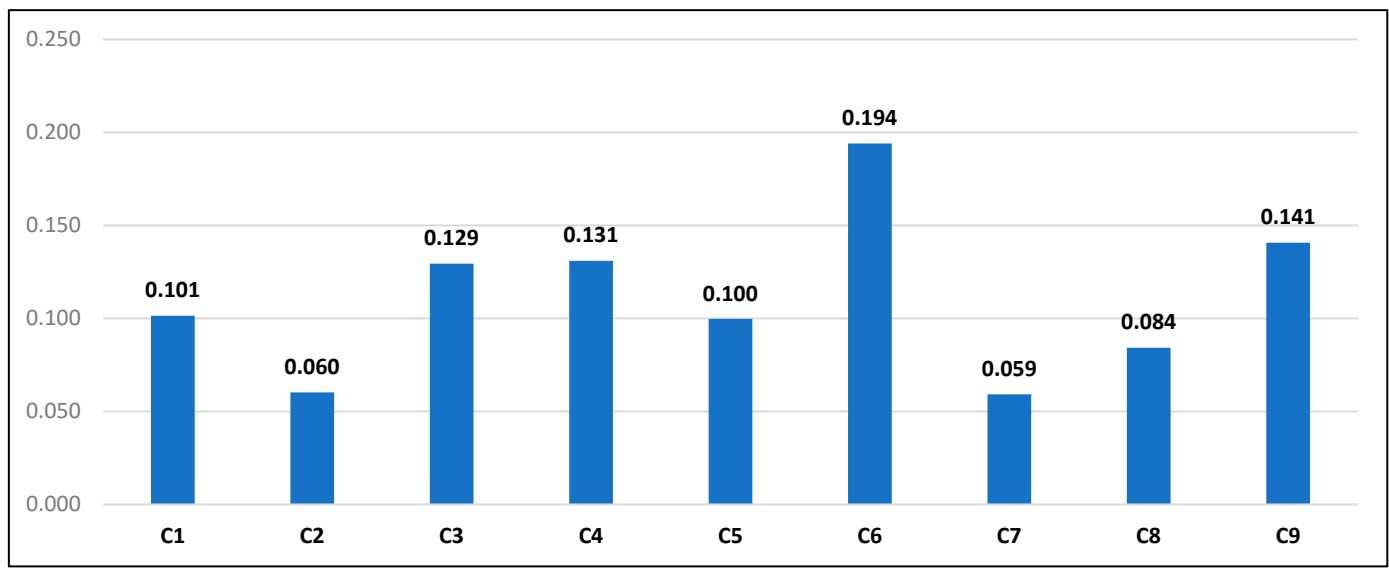

**Figure 8.** Probability of success $P_\nu^Y$ for candidates (for equally assigned SBJ criteria weights).

Moreover, Figure 10 serves as a visual summary and comparative diagram, providing insights into the objective importance $\mathcal{I}(Y|X = x_\mu)$, the subjective weighting $x_\mu^{SBJ}$, and the integrated importance $P_\mu^X \mathcal{S}(Y|X = X_\mu)$ of the criteria. This graphical presentation contributes to a more profound and holistic understanding of the interrelationships and distinctions among these factors within the context of personnel selection.

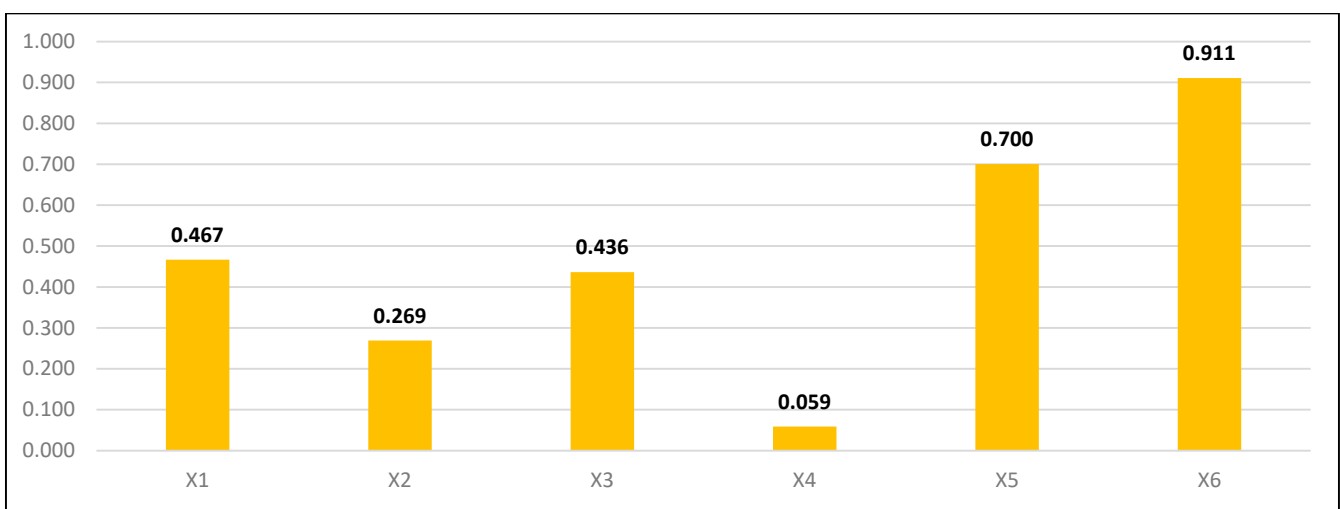

**Figure 9.** Integrated criteria importance $P_\mu^X \mathcal{S}(Y|X = X_\mu)$ produced by ES-MADM model (non-equally assigned SBJ criteria weights).

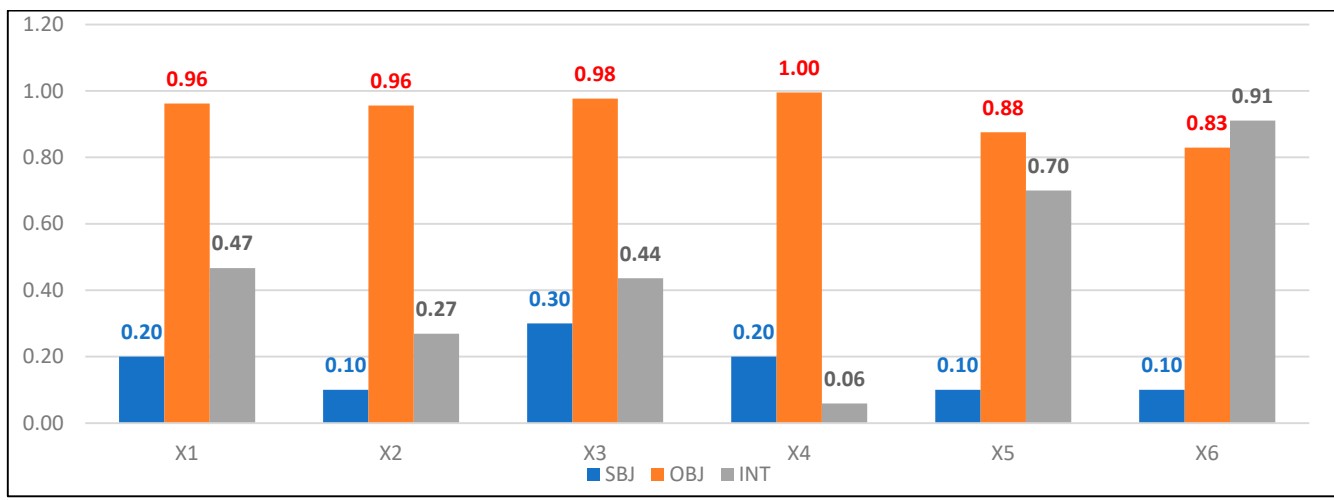

**Figure 10.** Comparison between the values of the integrated (INT), objective (OBJ), and subjective (SBJ) criteria importance (non-equally assigned SBJ criteria weights).

3.1.4. Sensitivity Analysis to ES-MADM Model

In order to scrutinize how the ES-MADM model responds to variations in the data matrix elements and pinpoint specific values that can impact the final outcome, a sensitivity analysis (SA) was conducted. This analysis aimed to evaluate how alterations in the data matrix elements influence the behavior of the model and identify values that could substantially affect the results.

The sensitivity analysis (SA) focused on the values associated with the second candidate, who, according to the findings presented in Table 8 and Figure 6, appeared to be the least preferable among the candidates. The objective was to maximize the preference probability for this candidate, making them the most preferable choice among all candidates. To perform the SA, the non-linear GRG (Generalized Reduced Gradient) solver, accessible through the EXCEL platform's toolbox, was employed. The solver was configured to consider the constraint that $\xi_{\mu 5}$ values must remain greater than 1 (minimum value) and less than 10 (maximum value).

The outcomes of the SA are concisely summarized in Tables 11 and 12, and these results are visually depicted in Figures 11 and 12. This rigorous analysis offers insights into how parameter variations impact candidate preference probabilities within the ES-MADM model.

**Table 9.** Computations summary in ES-MADM model for Steps 2–6 (non-equal SBJ criteria weights).

| Criteria | $P(Y=y_\nu\|X=x_\mu)=\xi_{\mu\nu}/\sum_{\nu=1}^{9}\xi_{\mu\nu}$ | | | | | | | | | $x_\mu^{INT}$ | $\mathcal{I}(Y\|X=x_\mu)$ | $P_\nu^Y=\sum_{\mu=1}^{6}P(Y=y_\nu\|X=x_\mu)\cdot P_\mu^X$ | | | | | | | | | $P_\mu^X\mathcal{S}(Y\|X=X_\mu)$ |
| --- | --- | --- | --- | --- | --- | --- | --- | --- | --- | --- | --- | --- | --- | --- | --- | --- | --- | --- | --- | --- | --- |
| | C1 | C2 | C3 | C4 | C5 | C6 | C7 | C8 | C9 | | (OBJ Criteria Importance) | C1 | C2 | C3 | C4 | C5 | C6 | C7 | C8 | C9 | (INT Criteria Importance) |
| X1 | 0.111 | 0.083 | 0.083 | 0.111 | 0.250 | 0.111 | 0.083 | 0.083 | 0.083 | 0.153 | 0.962 | 0.017 | 0.013 | 0.013 | 0.017 | 0.038 | 0.017 | 0.013 | 0.013 | 0.013 | 0.467 |
| X2 | 0.123 | 0.123 | 0.092 | 0.015 | 0.077 | 0.154 | 0.154 | 0.108 | 0.154 | 0.089 | 0.956 | 0.011 | 0.011 | 0.008 | 0.001 | 0.007 | 0.014 | 0.014 | 0.010 | 0.014 | 0.269 |
| X3 | 0.078 | 0.125 | 0.125 | 0.156 | 0.047 | 0.156 | 0.094 | 0.094 | 0.125 | 0.141 | 0.977 | 0.011 | 0.018 | 0.018 | 0.022 | 0.007 | 0.022 | 0.013 | 0.013 | 0.018 | 0.436 |
| X4 | 0.100 | 0.083 | 0.117 | 0.100 | 0.100 | 0.117 | 0.117 | 0.133 | 0.133 | 0.019 | 0.995 | 0.002 | 0.002 | 0.002 | 0.002 | 0.002 | 0.002 | 0.002 | 0.002 | 0.002 | 0.059 |
| X5 | 0.200 | 0.029 | 0.029 | 0.029 | 0.143 | 0.229 | 0.029 | 0.143 | 0.171 | 0.252 | 0.876 | 0.050 | 0.007 | 0.007 | 0.007 | 0.036 | 0.058 | 0.007 | 0.036 | 0.043 | 0.700 |
| X6 | 0.029 | 0.029 | 0.235 | 0.235 | 0.029 | 0.235 | 0.029 | 0.029 | 0.147 | 0.346 | 0.829 | 0.010 | 0.010 | 0.082 | 0.082 | 0.010 | 0.082 | 0.010 | 0.010 | 0.051 | 0.911 |

**Table 10.** Summary results table for ES-MADM model (non-equal SBJ criteria weights).

| Ranking of the Alternatives (Candidates) | | | | | | | | | Decision Stability | | |
| --- | --- | --- | --- | --- | --- | --- | --- | --- | --- | --- | --- |
| $P_{\nu=1}^Y$ | $P_{\nu=2}^Y$ | $P_{\nu=3}^Y$ | $P_{\nu=4}^Y$ | $P_{\nu=5}^Y$ | $P_{\nu=6}^Y$ | $P_{\nu=7}^Y$ | $P_{\nu=8}^Y$ | $P_{\nu=9}^Y$ | $\mathcal{S}(Y\|X)$ | $\mathcal{S}(Y)=-\sum_{\nu=1}^{N}P_\nu^Y\log_2 P_\nu^Y$ | $\mathcal{I}(Y\|X)=\frac{\mathcal{S}(Y\|X)}{\mathcal{S}(Y)}$ |
| 0.101 | 0.060 | 0.129 | 0.131 | 0.100 | 0.194 | 0.059 | 0.084 | 0.141 | 2.842 | 3.076 | 0.924 |

**Table 11.** Computations summary in ES-MADM model for Steps 2–6 (Sensitivity Analysis).

| Criteria | $P(Y=y_\nu\|X=x_\mu)=\xi_{\mu\nu}/\sum_{\nu=1}^{9}\xi_{\mu\nu}$ | | | | | | | | | $x_\mu^{INT}$ | $\mathcal{I}(Y\|X=x_\mu)$ | $P_\nu^Y=\sum_{\mu=1}^{26}P(Y=y_\nu\|X=x_\mu)\cdot P_\mu^X$ | | | | | | | | | $P_\mu^X\mathcal{S}(Y\|X=X_\mu)$ |
| --- | --- | --- | --- | --- | --- | --- | --- | --- | --- | --- | --- | --- | --- | --- | --- | --- | --- | --- | --- | --- | --- |
| | C1 | C2 | C3 | C4 | C5 | C6 | C7 | C8 | C9 | | (OBJ Criteria Importance) | C1 | C2 | C3 | C4 | C5 | C6 | C7 | C8 | C9 | (INT Criteria Importance) |
| X1 | 0.093 | 0.233 | 0.070 | 0.093 | 0.209 | 0.093 | 0.070 | 0.070 | 0.070 | 0.149 | 0.943 | 0.014 | 0.035 | 0.010 | 0.014 | 0.031 | 0.014 | 0.010 | 0.010 | 0.010 | 0.445 |
| X2 | 0.119 | 0.149 | 0.090 | 0.015 | 0.075 | 0.149 | 0.149 | 0.104 | 0.149 | 0.118 | 0.955 | 0.014 | 0.018 | 0.011 | 0.002 | 0.009 | 0.018 | 0.018 | 0.012 | 0.018 | 0.358 |
| X3 | 0.076 | 0.152 | 0.121 | 0.152 | 0.045 | 0.152 | 0.091 | 0.091 | 0.121 | 0.067 | 0.975 | 0.005 | 0.010 | 0.008 | 0.010 | 0.003 | 0.010 | 0.006 | 0.006 | 0.008 | 0.206 |
| X4 | 0.097 | 0.113 | 0.113 | 0.097 | 0.097 | 0.113 | 0.113 | 0.129 | 0.129 | 0.007 | 0.997 | 0.001 | 0.001 | 0.001 | 0.001 | 0.001 | 0.001 | 0.001 | 0.001 | 0.001 | 0.021 |
| X5 | 0.159 | 0.227 | 0.023 | 0.023 | 0.114 | 0.182 | 0.023 | 0.114 | 0.136 | 0.279 | 0.893 | 0.044 | 0.063 | 0.006 | 0.006 | 0.032 | 0.051 | 0.006 | 0.032 | 0.038 | 0.790 |
| X6 | 0.023 | 0.233 | 0.186 | 0.186 | 0.023 | 0.186 | 0.023 | 0.023 | 0.116 | 0.380 | 0.855 | 0.009 | 0.088 | 0.071 | 0.071 | 0.009 | 0.071 | 0.009 | 0.009 | 0.044 | 1.031 |

**Table 12.** Summary results table for ES-MADM model (Sensitivity Analysis).

| Ranking of the Alternatives (Candidates) | | | | | | | | | | Decision Stability | | |
|---|---|---|---|---|---|---|---|---|---|---|---|---|
| $P^Y_{v=1}$ | $P^Y_{v=2}$ | $P^Y_{v=3}$ | $P^Y_{v=4}$ | $P^Y_{v=5}$ | $P^Y_{v=6}$ | $P^Y_{v=7}$ | $P^Y_{v=8}$ | $P^Y_{v=9}$ | $\mathcal{S}(Y|X)$ | $\mathcal{S}(Y)=-\sum\limits_{v=1}^{N} P^Y_v \log_2 P^Y_v$ | $\mathcal{I}(Y|X)=\frac{\mathcal{S}(Y|X)}{\mathcal{S}(Y)}$ |
| 0.087 | 0.215 | 0.107 | 0.103 | 0.084 | 0.164 | 0.050 | 0.070 | 0.119 | 2.851 | 3.046 | 0.936 |

**Revised Values for the 2nd Candidate to Maximize Selection Probability $P^Y_{v=2}$**

| $X_1$ | | $X_2$ | | $X_3$ | $X_4$ | $X_5$ | $X_6$ |
|---|---|---|---|---|---|---|---|
| 10 | | 10 | | 10 | 7 | 10 | 10 |

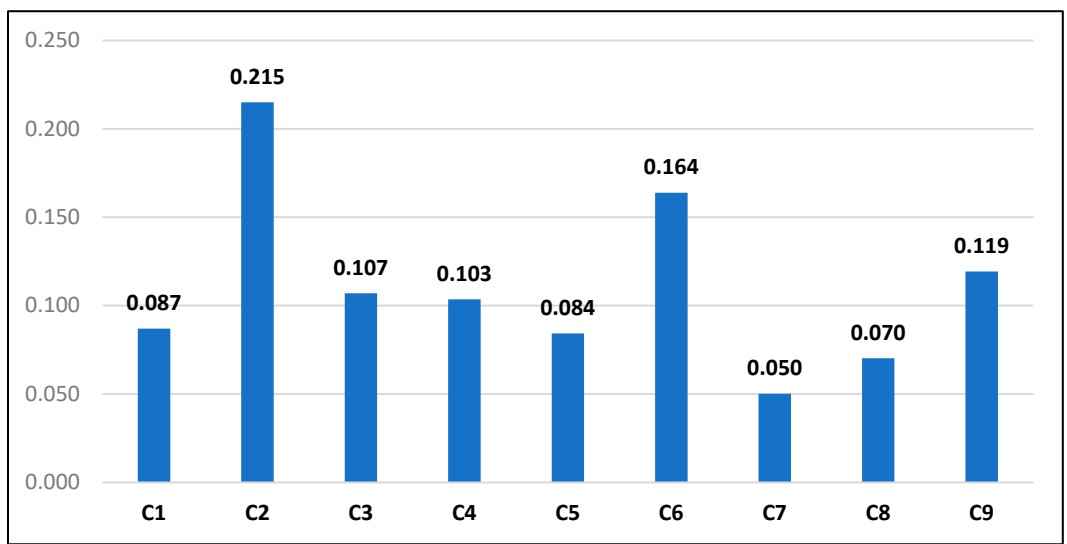

**Figure 11.** Probability of selection $P_\nu^Y$ for candidates (SA Results $\xi_{\mu 2} - P_2^Y$).

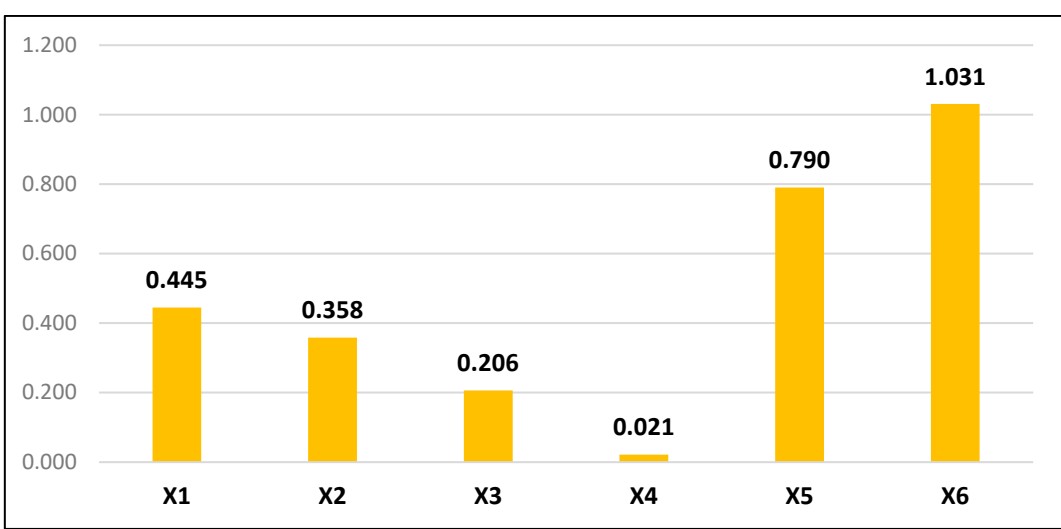

**Figure 12.** Integrated criteria importance $P_\mu^X \mathcal{S}(Y|X = X_\mu)$ produced by ES-MADM model (SA Results $\xi_{\mu 2} - P_2^Y$).

## 4. Results

### 4.1. ES-MADM Model: An Overview to the Results

The conducted case study delves into the intricacies of personnel evaluation and selection for a specific organizational role. In particular, the ES-MADM model processes data associated with the predefined criteria employed for personnel assessment. It delineates the selection probabilities for each candidate, representing their respective preference levels. Furthermore, the ES-MADM model conducts a comprehensive evaluation of the criteria utilized in the assessment process, elucidating their collective significance concerning the final outcomes. Additionally, the model furnishes a valuable mathematical tool for appraising the stability of the decision problem. This assessment pertains to the overarching relationship between the established criteria and the available alternatives, affording decision makers profound insights into the stability of the decision-making process.

### 4.2. Case Study (Initial Results)

The initial dataset of the case study involved the equitable distribution of subjective weights across the criteria. Tables 6 and 7 meticulously present a comprehensive

compilation of the outcomes derived from the ES-MADM model. These results hold pivotal significance as they underpin the assessment process. The consolidated findings are concisely summarized in Table 13, which offers valuable and discerning insights.

**Table 13.** Summary table for ES-MADM model results (first case study).

| Candidate Importance ($P_v^Y$) | | Most Important Criterion (INT) $X_6$ | Least Important Criterion (INT) $X_4$ | Problem Stability $\mathcal{I}(Y|X)$ |
|---|---|---|---|---|
| **Most Important** | **Less Important** | | | |
| C6 | C2 | 1.110 (Recommendation Letters) | 0.036 (Proficiency in MS OFFICE) | 0.917 |

### 4.2.1. Results Analysis

An analysis of the findings presented in Table 13 underscores the pronounced superiority of candidate C6, with candidate C2 appearing to be the least influential. Furthermore, the criterion $X_6$ (Recommendation Letters) emerges as the predominant factor, while the criterion $X_4$ (Proficiency in MS OFFICE) appears to exert minimal influence. This observation aligns with the observation that all candidates exhibit relatively similar performance levels in using MS OFFICE platforms, whereas in the case of criterion $X_6$, there are substantial disparities among the candidates, leading to markedly divergent outcomes. Regarding the stability of the problem, it can be asserted that it is relatively low, as indicated by the proximity of the objective importance measure $\mathcal{I}(Y|X)$ to unity. This suggests a weak connection or correlation between the criteria ($X$) and the outcome ($Y$). In other words, the criteria alone may not be robust predictors of the selection process, implying that additional factors may play a significant role in candidate selection.

### 4.2.2. Comparison of the Results Produced by ES-MADM with Those from TOPSIS

Upon comparing the outcomes generated by the ES-MADM model with those obtained from the TOPSIS model, as utilized by Oya KORKMAZ [34], we arrive at the comparative diagram showcased in Figure 13.

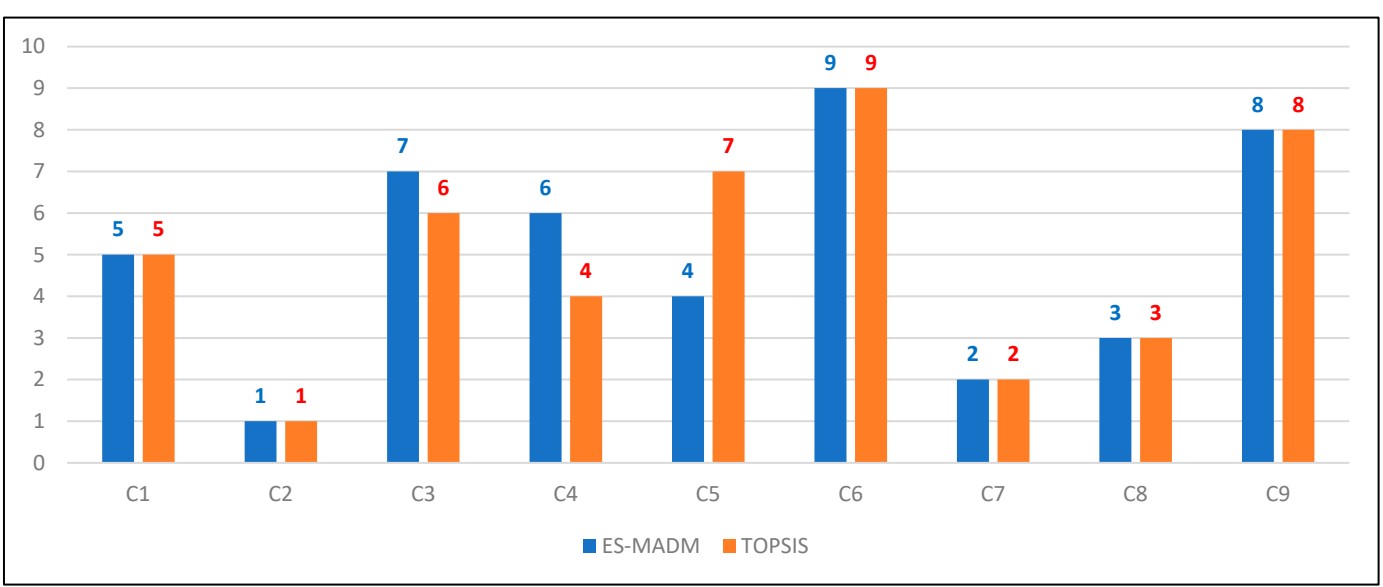

**Figure 13.** Comparative analysis of candidate ranking: ES-MADM model vs. TOPSIS.

This illustration distinctly reveals a convergence in results, particularly in the identification of the best- and worst-performing candidates, namely C6 and C2, respectively. Remarkably, a consensus emerges in the outcomes for candidates C1, C2, C3, C7, C8, and

C9, where the two models produce congruent results. Nevertheless, slight disparities in the ranking are observed for candidates C3, C4, and C5.

This phenomenon can be attributed to the inherent variations in the methodologies and underlying assumptions of the two models. The alignment of results for certain candidates signifies a harmonious interpretation of their performance.

In contrast, the minor differences in the ranking of other candidates may stem from nuanced differences in how the models weigh and evaluate specific criteria or candidate attributes. These distinctions underscore the importance of considering the models' unique characteristics and the specific context in which they are applied to make informed decisions.

*4.3. Case Study (Assigning Non-Equal Subjective Weights to the Criteria)*

The ES-MADM model demonstrates a unique capacity to delve into the inherent information embedded within the data matrix. This analytical prowess empowers the model to compute objective weights, which can then be harmonized with the subjective criteria weights determined by decision maker(s). The end result is a harmonized set of integrated criteria weights, fusing both objective and subjective evaluations. This integrated approach imparts a higher degree of solidity, reliability, and precision to the criteria weights.

Crucially, this phase of the analysis assumes that decision makers ascribe non-uniform subjective weights to the criteria. Consequently, the model amalgamates these subjective judgments with the objective weights it calculates autonomously. This amalgamation signifies that the model relies on the provided data and seamlessly combines them with the subjective assessments of decision makers, ultimately generating results that are both data-driven and reflective of decision maker preferences.

To evaluate the impact of subjective criteria weights on the overarching results, encompassing selection probabilities, criteria significance, and problem stability, a comparative analysis was conducted between scenarios with equal and non-equal subjective criteria weights. The objective of this comparative examination is to elucidate the disparities that emerge when subjective weights are considered or omitted.

The consolidated findings of this comparative analysis are succinctly presented in Table 14, furnishing a lucid and systematic summary of how subjective criteria weights influence the comprehensive results. This analytical approach offers valuable insights into the implications of subjective weight variations in the decision-making process.

**Table 14.** Comparison table for ES-MADM results (equal and non-equal SBJ criteria weights).

| Subjective Criteria Weights | Candidate Importance ($P_{\nu}^{Y}$) | | Most Important Criterion (INT) | Least Important Criterion (INT) | Problem Stability $\mathcal{I}(Y|X)$ |
|---|---|---|---|---|---|
| | Most Important | Most Important | | | |
| Non-equal | C6 | C7 | $X_6$ | $X_4$ | 0.924 |
| Equal | C6 | C2 | $X_6$ | $X_4$ | 0.917 |

The conclusive findings presented in Table 14 demonstrate the implications of assigning non-equal subjective weights to the criteria. These findings can be summarized as follows:

- **Candidate Importance:** The selection probabilities pertaining to the candidates, denoting their relative importance, undergo variations, particularly concerning the least significant candidate. In cases with non-uniform subjective criteria weights, candidate C7 ($Y_7$) emerges as the least favored option, exhibiting the lowest selection probabilities. Conversely, candidate C2 ($Y_2$) consistently maintains the highest-ranking score across both scenarios. This can be attributed to candidate C2 consistently displaying superior values across all criteria in comparison to the other candidates.

- **Integrated Criteria Significance:** In both scenarios, it is evident that criterion $X_6$, specifically pertaining to "Recommendation Letters," consistently emerges as the most pivotal, while $X_4$ is consistently identified as the least influential. This observation

aligns seamlessly with the fact that for both scenarios, all candidates exhibit relatively uniform performance levels concerning criterion $\mathbf{X}_4$, specifically their usage of MS OFFICE platforms. In stark contrast, when considering criterion $\mathbf{X}_6$, significant disparities among the candidates become apparent, resulting in notably divergent outcomes. To elucidate further, when there is substantial divergence in values among the candidates for a particular criterion, it leads to lower entropy measures. Consequently, this reinforces the credibility of that criterion in explaining and contributing to the decision-making process, hence ascribing it higher importance. In essence, the degree of variation in candidate values for a given criterion directly impacts its reliability and influence in guiding the decision, resulting in its elevated importance.

- **Problem Stability:** The stability of the problem seems to diminish when non-uniform subjective criteria weights are introduced, exemplified by the elevated value of $\mathcal{I}(\mathbf{Y}|\mathbf{X})$ ($\mathcal{I}(\mathbf{Y}|\mathbf{X}) = 0.924$) in the second scenario. This phenomenon suggests a reduction in the reliability and consistency of the decision-making process. This trend occurs because when non-uniform subjective weights are assigned to criteria, there is a greater divergence in the importance attributed to various factors by decision makers. Consequently, the criteria may exert more variable influences on the final outcome, resulting in a less stable decision-making environment. The increased value of $I(Y|X)$ in the second scenario underscores the greater uncertainty and variability introduced by non-uniform subjective weights, which can pose challenges in achieving consistent and reliable decisions.

- **Criteria Importance:** Figures 7 and 10 offer a comparative visualization of the subjective importance $\mathbf{P}_{\mu}^{\mathbf{X}}$, the objective importance $\mathcal{I}(\mathbf{Y}|\mathbf{X} = \mathbf{x}_{\mu})$ and the integrated importance $\mathbf{P}_{\mu}^{\mathbf{X}}\mathcal{S}(\mathbf{Y}|\mathbf{X} = \mathbf{X}_{\mu})$ of the criteria. A clear distinction emerges when considering the scenario of equal subjective criteria weights versus the one where these weights differ significantly. In the case of equal subjective criteria weights, the integrated significance of the criteria primarily hinges on the objective weights, which are derived from the data matrix. These objective weights essentially constitute the primary contributors to the overall assessment of criteria significance. Conversely, when subjective criteria weights exhibit substantial variability, a different dynamic emerges. In this scenario, the objective-derived criteria weights, along with their corresponding objective significance measures, play a critical role as correctors. They refine and recalibrate the subjective assessments by introducing an objective perspective, grounded in the inherent information encapsulated within the data matrix. This comparison, as depicted in Figure 10, underscores the significance of integrating objective weights and their associated significance measures into the evaluation process. These objective elements serve as essential correctors for subjective evaluations, ensuring a more precise, equitable, and comprehensive appraisal of the overall criteria significance. Their inclusion helps strike a balance between subjective judgments and data-driven objectivity, leading to more informed and robust decision-making processes.

*4.4. Sensitivity Analysis Results*

The sensitivity analysis conducted aimed to assess the model's capability to identify the specific conditions within the data matrix (values of $\xi_{\mu\nu}$) that would lead to a favorable alteration of the selection probabilities ($\mathbf{P}_{\nu}^{\mathbf{Y}}$) for a certain candidate. In particular, the analysis focused on determining the necessary adjustments to the second candidate C2 ($\xi_{\mu 2}$) to achieve this desired outcome. The comprehensive results of this analysis are summarized in Table 15, providing a structured and concrete representation of the findings. Additionally, Table 16 provides a concise summary of the variations or differentiations observed between the initial data and the results derived from the sensitivity analysis test.

**Table 15.** Summary table for ES-MADM model results (sensitivity analysis findings).

| Candidate Importance ($P_v^Y$)-Ranking | | Most Important Criterion (INT) | | Least Important Criterion (INT) | | Problem Stability $\mathcal{I}(Y\|X)$ | |
|---|---|---|---|---|---|---|---|
| Before SA | After SA | Before SA | After SA | Before SA | After SA | Before SA | After SA |
| 0.050 9th favorable | 0.215 1st favorable | $X_6$ | $X_6$ | $X_4$ | $X_4$ | **0.917** | **0.936** |

**Table 16.** SA results with respect to C2 Characteristics to achieve maximum $P_2^Y$.

| Criteria | | Second Candidate | | |
|---|---|---|---|---|
| | | **Before SA** | **After SA** | **Percentage** |
| X1 | Logistics Experience | 3 | 10 | 233% |
| X2 | Education | 8 | 10 | 25% |
| X3 | Flexible Working Hours and Overtime | 8 | 10 | 25% |
| X4 | Proficiency in MS Office Programs | 5 | 7 | 40% |
| X5 | Package Software Used in The Field of Logistics | 1 | 10 | 900% |
| X6 | Recommendation Letters | 1 | 10 | 900% |
| Average Percentage of 2nd Candidate Increase in Criteria Values | | | | **354%** |

The conclusive findings, as presented in Table 16, unmistakably emphasize the substantial requirements for the second candidate to attain the maximum selection probability score, $P_2^Y$, thereby shifting the selection probabilities in their favor. This necessitates a significant reinforcement across multiple criteria. On average, an approximately 354% enhancement is deemed necessary to achieve the desired outcome. It is important to note that there are no changes in the most and least important criteria, $X_6$ and $X_4$, respectively. This phenomenon is attributed to the diversification rate for these criteria, which has been elucidated in Section 4.3.

This analysis underscores the remarkable capability of the ES-MADM model, seamlessly integrated within the EXCEL platform, to pinpoint the specific values of particular military capabilities (criteria) that necessitate reinforcement to yield favorable winning probabilities. This feat was accomplished through the utilization of the non-linear GRG (Generalized Reduced Gradient) solver, a powerful tool that optimizes criteria values to achieve desired outcomes effectively. In summation, this analysis serves as a tangible demonstration of the practical utility of the ES-MADM model. It showcases the model's ability to offer valuable insights by identifying the precise enhancements required in military capabilities to secure favorable winning probabilities, thereby contributing to informed and strategic decision-making processes.

## 5. Discussion

The proposed ES-MADM model serves as a sophisticated computational framework tailored to empower decision makers involved in personnel selection processes. It is designed to cater to the specific needs of organizations and firms striving to make informed choices when selecting candidates for various positions.

The ES-MADM model seamlessly integrates objective data with subjective assessments, allowing decision makers to derive precise and well-informed criteria weights. This integration enhances the reliability and accuracy of personnel selection outcomes, ensuring that the chosen candidates align with the organization's requirements.

An essential feature of the ES-MADM model is its ability to assess the significance of different criteria. By doing so, it offers valuable insights into the influence of various factors on the selection process. This analysis helps in understanding the dependencies between criteria and aids in evaluating decision stability.

The model is represented in a clear and structured flowchart, as depicted in Figures 2 and 3. This visual representation streamlines the decision-making process and is partic-

ularly beneficial when prompt and accurate decisions are paramount in the personnel selection context.

Furthermore, the ES-MADM model's versatility extends beyond personnel selection and can be applied effectively in various decision scenarios. It excels in offering insights that reveal the strengths and weaknesses of candidates, ultimately aiding in the selection of the most suitable individuals for specific roles.

In the context of advancing the ES-MADM model, potential improvements include integrating additional decision-making principles and exploring the use of fuzzy set inputs for the data matrix to address vagueness in candidate information. However, the model currently has limitations that could be addressed in future work. Notably, it does not account for interlinkages between criteria, neglecting their potential interaction and influence. Additionally, the absence of consideration for fuzziness in the values of the data matrix poses a constraint. Investigating these aspects in subsequent research endeavors holds promise for refining the model and enhancing its practical applicability.

Furthermore, the integration of the CRITIC method [35] holds the potential to shed light on correlations between different criteria, offering valuable insights into the complex interrelationships among various factors. Such an inclusion would contribute to a more comprehensive understanding of the decision problem.

In forthcoming research, the exploration of integrating quantum X-entropy [36], Evidential Fuzzy Multicriteria Decision Making (EFMCDM) based on belief entropy [37], and the Maximum Entropy Negation of a Complex-Valued Distribution [38] into the ES-MADM model could offer a valuable avenue for addressing potential limitations of the existing model in the context of personnel selection. Investigating the synergies between these advanced methodologies may provide enhanced capabilities for handling uncertainty and refining decision-making processes, thereby contributing to the advancement and optimization of the ES-MADM model for practical applications in personnel selection scenarios.

To bolster the model's analytical prowess, conducting comparative analyses against a spectrum of alternative MADM methods, including but not limited to the Analytical Hierarchy Process (AHP) [39], the Analytical Network Process (ANP) [40], ELECTRE [41], PROMETHEE [3] and other MCDM methods, presents an avenue for gaining valuable insights into the nuances of the personnel selection process.

In summation, the ES-MADM model, with the prospect of these future enhancements, emerges as a promising asset in the realm of personnel selection. By refining the decision-making process within this domain, it stands ready to assist organizations in making judicious, well-informed, and dependable choices when selecting candidates for various positions.

## 6. Conclusions

In summary the ES-MADM (Entropy-based Multi-Criteria Decision Making) model is a powerful tool for personnel selection. It seamlessly integrates objective and subjective data, enhancing the accuracy of candidate assessments. The model's ability to evaluate criteria significance, assess decision stability, and provide clear visual representations streamlines decision-making. Future improvements, such as incorporating additional methodologies and accommodating data vagueness, hold promise. Comparative analyses with other decision-making methods can further enrich the selection process. Overall, the ES-MADM model offers a sophisticated solution for making well-informed and precise candidate selections, with the potential to adapt to various decision contexts.

**Author Contributions:** S.K.: writing and editing, V.T.: review and supervision. All authors have read and agreed to the published version of the manuscript.

**Funding:** This research received no external funding.

**Data Availability Statement:** The data presented in this study are available on https://dergipark. org.tr/tr/download/article-file/611012 (accessed on 1 November 2023) and https://github.com/ skiratsoudis/Personnel-Selection-Problem--ES-MADM-Model (accessed on 1 November 2023).

**Conflicts of Interest:** The authors declare no conflict of interest.

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
