# Peer review of "Enhancing Personnel Selection through the Integration of the Entropy Synergy Analysis of Multi-Attribute Decision Making Model: A Novel Approach"

_information, doi:10.3390/info15010001_

Round 1

Reviewer 1 Report

Comments and Suggestions for Authors

This paper presents an innovative model called Entropy Synergy Analysis of Multi-Attribute Decision Making (ES-MADM). The importance of model evaluation criteria and sensitivity analysis were conducted to verify the overall effectiveness of the ES-MADM model through experiments. I don't think it can be published in its current form. I provide some suggestions below for the revised version:

1. In Page 4, after the sentence “In their academic work,”, there is an extra space. Please modify it.

2. In Page 6, there is an extra comma after “where” in Equation 1.

3. In Section 2.2.4, the sentence "mutual information is the measures the amount of information that can be obtained about one random variable by observing another" is not easy to understand. Please modify it to make it easier to read.

4. The format of the pictures and tables in the article is not suitable. Please modify it to make the overall style of the paper consistent.

5. What are the limitations of this approach?

6. Some current related works are suggested to discuss, e.g., Quantum X-entropy in generalized quantum evidence theory; EFMCDM: Evidential Fuzzy Multicriteria Decision Making Based on Belief Entropy; On the maximum entropy negation of a complex-valued distribution.

7. Some pictures are not clear enough. Please modify the pictures to make them clearer.

Comments on the Quality of English Language

Minor editing of English language required. 

Author Response

Response to the comments of the 1st Reviewer,

 Page 4, Extra Space:

The additional space after the sentence "In their academic work" on Page 4 has been duly rectified.

  1. Page 6, Extra Comma:

The extraneous comma after "where" in Equation 1 on Page 6 has been removed.

  1. Section 2.2.4, Sentence Clarity:

The sentence regarding mutual information in Section 2.2.4 has been rephrased to enhance readability.

  1. Format of Pictures and Tables:

The format of pictures and tables has been modified for improved consistency throughout the paper. Initial Figures 2 and 3 have been enhanced and replaced, while Figures 4 and 5 have been substituted with Tables 3 and 4, respectively.

  1. Limitations of the Approach:

A new paragraph discussing the limitations of the proposed method has been added in the discussion section.

  1. Inclusion of Related Works:

Pertinent related works, such as Quantum X-entropy and EFMCDM, have been incorporated into the discussion. Additionally, a paragraph on potential future works, involving the combination of the ES-MADM model with these methods to address vagueness in personnel selection problems, has been included.

  1. Picture Clarity:

To enhance visual clarity, we have made specific adjustments to graphical elements. Figures 2, 3 and 6 have been refined, while Tables 3 and 4 now replace the original Figures 4 and 5.

Reviewer 2 Report

Comments and Suggestions for Authors

This manuscript presents a novel multi-attribute decision making model, which incorporates both subjective and objective criteria into a unifying framework. The proposed approach employs Information Theory concepts to qualitatively and quantitively calculate a plethora of crucial factors that affect decision-making policies, such as the significance of the criteria, the total uncertainty of the process and the ranking of each alternative. The authors illustrate the effectiveness of their model via a personnel selection task where there are 9 different candidates and 6 different criteria.

The authors should consider re-structuring the majority of their manuscript as they have included the proposed model and the experiments in the same section. They are highly advised to include their experimental setup and their results in a different section so that the reader can easily locate the different aspects of the paper. By adopting the above recommendation, the authors should also limit the number of the sub-sections.

As far as related approaches are concerned, authors fail to include a series of recent works on personnel selection that are based on Machine Learning. Representative examples of such works that deserve to be mentioned in this paper include:

·       Goretzko, D., & Israel, L. S. F. (2021). Pitfalls of machine learning-based Personnel Selection. Journal of Personnel Psychology.

·       Kanakaris, N., Giarelis, N., Siachos, I., & Karacapilidis, N. (2022). Making personnel selection smarter through word embeddings: A graph-based approach. Machine Learning with Applications7, 100214.

·       Kanakaris, N., Giarelis, N., Siachos, I., & Karacapilidis, N. (2021). Shall I Work with Them? A Knowledge Graph-Based Approach for Predicting Future Research Collaborations. Entropy23(6), 664.

·       König, C. J., & Langer, M. (2022). Machine learning in personnel selection. Handbook of Research on Artificial Intelligence in Human Resource Management, 149-167.

·       Zhang, N., Wang, M., Xu, H., Koenig, N., Hickman, L., Kuruzovich, J., ... & Kim, Y. Reducing subgroup differences in personnel selection through the application of machine learning. Personnel Psychology.

The authors present the proposed model in a detailed fashion, providing the necessary equations describing each component of their approach as well as figures and tables that summarize the overall architecture. While the model’s complexity introduce a series of challenges, the authors manage to explain each step in rigorously. They are, however, advised to clarify the significance of each concept they utilize and how it is exploited as a unique insight to the multi-attribute decision-making process.

The experiments conducted are easily comprehensible and reproducible and reflect the effectiveness of the proposed approach. Nevertheless, the authors could have compared their approach with other models. Such a practice could help readers to distinguish the advantages and the limitations of the proposed approach.

Finally, we recommend authors to include a link for the datasets and code of their approach and experiments, hosted in a public code repository (e.g., GitHub), to further improve the reproducibility of their work.

Author Response

Response to 2nd Reviewer:

  1. Restructuring of Manuscript:

The manuscript has been restructured by segregating the model description and case studies-sensitivity analysis section.

  1. Inclusion of Additional Related Works:

Representative recent works on personnel selection based on Machine Learning, as suggested, have been included in the introduction section.

  1. Clarification of Model's Significance:

The significance of each mathematical concept utilized in the model has been elucidated in the new Tables 3 and 4.

  1. Model Comparison:

The proposed approach has been compared with the TOPSIS model, and the results are thoroughly discussed in paragraph 4.2.2.

  1. Provision of Datasets and Code:

The datasets are made available at https://dergipark.org.tr/tr/download/article-file/611012 and https://github.com/skiratsoudis/Personnel-Selection-Problem--ES-MADM-Model, enhancing the reproducibility of the experiments.

Reviewer 3 Report

Comments and Suggestions for Authors

Dear Authors:

I have gone through the manuscript. It is well written. My only concern is that you covered research till 2011. I suggest including the latest research {from 2011 to 2023]. Literature review is therefore not exhaustive.

Second, introduction is too long. You can reduce it to two paragraphs. Bring the essence of what you mentioned in four pages can be said in two paragraphs.

Other things are fine. 

Good luck.

Author Response

  1. Response to 3rd Reviewer:

Response to 3rd Reviewer:

  1. Inclusion of Latest Research:

Recent works on personnel selection have been added, spanning from 2011 to 2023, addressing the suggestion for a more exhaustive literature review.

  1. Reduction of Introduction Length:

The introduction section has been meticulously revised and condensed to two paragraphs, preserving the essence of the information presented.

Round 2

Reviewer 2 Report

Comments and Suggestions for Authors

All issues raised on the initial version of the manuscript have been adequately addressed.